# Design Strategies for Electrochemical Aptasensors for Cancer Diagnostic Devices

**DOI:** 10.3390/s21030736

**Published:** 2021-01-22

**Authors:** Kamila Malecka, Edyta Mikuła, Elena E. Ferapontova

**Affiliations:** 1Institute of Animal Reproduction and Food Research, Polish Academy of Sciences, Tuwima Str. 10, 10-748 Olsztyn, Poland; k.malecka@pan.olsztyn.pl (K.M.); e.mikula@pan.olsztyn.pl (E.M.); 2Interdisciplinary Nanoscience Center (iNANO), Faculty of Science and Technology, Aarhus University, Gustav Wieds Vej 14, 8000 Aarhus C, Denmark

**Keywords:** aptasensor, aptamer, cancer biomarkers, electrochemical biosensors

## Abstract

Improved outcomes for many types of cancer achieved during recent years is due, among other factors, to the earlier detection of tumours and the greater availability of screening tests. With this, non-invasive, fast and accurate diagnostic devices for cancer diagnosis strongly improve the quality of healthcare by delivering screening results in the most cost-effective and safe way. Biosensors for cancer diagnostics exploiting aptamers offer several important advantages over traditional antibodies-based assays, such as the in-vitro aptamer production, their inexpensive and easy chemical synthesis and modification, and excellent thermal stability. On the other hand, electrochemical biosensing approaches allow sensitive, accurate and inexpensive way of sensing, due to the rapid detection with lower costs, smaller equipment size and lower power requirements. This review presents an up-to-date assessment of the recent design strategies and analytical performance of the electrochemical aptamer-based biosensors for cancer diagnosis and their future perspectives in cancer diagnostics.

## 1. Introduction

Cancer is a high-mortality chronic disease and a serious public health problem. With ageing of the population, it becomes more and more common around the world. One in three can be expected to be diagnosed with cancer in our lifetimes and one in four will die of it. In 2018, 18.1 million people worldwide had cancer and 9.6 million died from the disease. By 2040, these numbers will be almost doubled, with the greatest increase in low- and middle-income countries [1]. 

During last decades, a pronounced improvement in cancer diagnosis and treatment has been achieved: people suffering from some of the most common forms of cancers are twice as likely to survive for at least 10 years, compared with patients diagnosed in the early 1970s [2,3]. Tremendous progress in survival ratios has been achieved with some types of cancer, which, among other factors, can be attributed to the early detection of tumours and the greater availability of screening tests. However, for certain types of cancer the survival ratios are still low (Figure 1). Most of those patients enter the healthcare system through the portal of a pathology diagnosis: most primary and recurrent diagnoses of cancer are at present based on the pathology tissue diagnoses [4]; that also means that most cancers are diagnosed too late or are misdiagnosed, which prevents successful treatment. The five-year survival rate for cancer of the pancreas is just 6%; for oesophageal cancer it is 13%; and for lung cancer it is 16%, which can be related to poorer early diagnosis of tumours [5] (Figure 1). Aggressive types of breast cancer, oesophagus, and liver cancers are also characterised by low five-year-survival rates, less than 20% [6], and more advanced precision medicine approaches, including early-stage and precise diagnosis of patient’s specific cancer, are required for their successful treatment.

It is therefore clear that one of the most important factors in the fight against cancer is its early and reliable diagnosis: any cancer is easier to treat when the treatment is started early. The efforts are concentrated on the development of robust and non-invasive tools for early diagnosis and prognosis of cancer and its timely therapy and treatment monitoring. Liquid biopsy assays allow such a non-invasive assessment of specific tumour biomarkers via a conventional blood draw [8], and sensitive and robust biomarkers for liquid biopsy-based tumour diagnosis and efficient methods for their analysis are currently one of the most challenging problems in cancer diagnosis research.

The molecular recognition elements capable of strong and selective binding of such cancer biomarkers as blood-, urine-, and saliva-circulating proteins are the most important components of such assays, providing both their high specificity and sensitivity. Among those, nucleic acid and peptide aptamers are highly attractive molecular recognition elements that can bind their targets, from small molecules to cells, with high affinity and specificity [9,10]. Unlike antibodies, DNA and RNA aptamers can be produced in vitro by the procedure termed Systematic Evolution of Ligands by Exponential Enrichment (SELEX) [11]. Due to their nucleic acid nature, they are amenable to chemical synthesis, which facilitates their production and extends their modification possibilities [12]. For successful operation in biological matrices, the aptamer selection is also performed in the natural environment in which they are planned to be used, in order to ensure the optimum binding properties of the aptamers in clinical samples [13]. The aptamers can be also readily engineered into bi-specific dimer aptamers [14,15]. Construction of “multivalent” aptamers, dimers in their simplest form, capable of binding to multiple protein binding sites, essentially improves the biorecognition and affinity properties of the aptamers. Of particular advantage for their use in sensors is that aptamers can be easily modified for any surface immobilisation and engineered to be highly stable and active [16,17] (Figure 2). 

Peptide aptamers emerged later than the nucleic acid aptamers, pioneered by thioredoxin A (TrxA) affinity protein reported in 1996 [10] and defined, by analogy with nucleic acid aptamers, as a “peptide aptamer.” The peptide aptamers are chosen from combinatorial libraries for their protein affinity properties and may be produced by expression in bacterial cells [21,22]. Hitherto, the peptide aptamers are rarely used in cancer diagnostic devices and thus are excluded from the scope of this review. Further the term “aptamer’’ will be used here only in connection with the nucleic acids aptamers, except for one example of an aptasensor for vascular endothelial growth factor detection based on a peptide aptamer [23].

As alternatives to antibodies, aptamers are currently intensively used in design of novel biosensing principles [12]. Their advantageous characteristics such as higher or comparable target affinity, long-term stability, lower batch-to-batch variation, lower preparation cost and low immunogenicity [24,25] make them most attractive biorecognition molecules in the biosensor development [26]. They can also excellently distinguish between different modified forms and isoforms of the same protein, and their affinities can be finely tuned by manipulating their binding reaction conditions and their sequence compositions [27]. Because of the convenience of their chemical modification, conjugation and redox labelling, and controlled immobilisation [27,28,29], the aptamers represent almost the ideal biorecognition molecules for the electrochemical biosensor design [12,30,31,32]. The main trends and recent advances in the electrochemical aptasensor research for cancer diagnosis are scrutinised and critically discussed in this review. 

## 2. Electrochemical Techniques

In the electrochemical biosensor approach scrutinised further, the electrode is a transducer element. Electrical changes induced by the protein binding to the biorecognition interface can result either from reduction or oxidation reactions of a redox marker or due to interfacial changes and thus are analysed in several ways, such as via measuring the current or potential responses when processes involving production or consumption of electrons are involved. When such changes are not caused by direct electron flow, the measured responses are resistance, capacitance, or impedance. Generally, converting a biological interaction into an electrical signal is straightforward and can be measured and quantified by a variety of methods such as potentiometry, amperometry, voltammetry, conductometry, and impedance [33].

In chronoamperometry, most commonly used in biosensors, direct current is measured by applying a constant potential to the bio-modified working electrode. The amperometric response changes after the analyte binding and that allows to quantify it [34]; the response is produced either by the redox indicator or the analyte itself, if it is electroactive or conditionally can undergo a redox transformation [34]. The current response is a measure of the electron transfer rate and is proportional to the concentration of the analyte [35,36].

Voltammetric methods, such as cyclic (CV), pulse differential (DPV), alternating current (ACV), linear (LSV), and stripping voltammetry, etc., are other mainstream techniques used in biosensors due to their high sensitivity, precision, accuracy, and informativity; they relate the current response from the redox markers to the potential applied [37]. The redox peak potentials are specific to the analysed system, and the magnitude of the peak current is proportional to the concentration of the analyte both if it is redox-active or if the proper redox markers are used. Voltammetric techniques are versatile and allow easy extraction of characteristic information about the analyte. CV is regularly used for determination of formal potentials, redox process mechanisms, and electron transfer kinetics [38]. The DPV and square wave voltammetry (SWV) detections have become widely applied in biosensors in recent years due to their higher sensitivity, and, as a result, selectivity. In particular, SWV is often used in fast analytical protocols, due to its ability to operate at high frequencies [39], which can also minimise the consumption of electroactive species in comparison to other pulse techniques [40]. The square wave frequency is a parameter that arises from the application of the square wave on the staircase potential and is the frequency at which the analyte is sampled. Similar to CV, the increment in the SWV sweep rate also correlates with an increase in the peak current, however, unlike CV, this will be proportional to the logarithm of the square wave frequency [41].

Electrochemical impedance spectroscopy (EIS) is another technique heavily exploited in biosensor research, and most productive in detecting the interfacial changes of electrodes functionalised with a biological material [42]. In EIS, a sinusoidal voltage is applied and the resulting current is measured. Impedance is then calculated as a ratio of voltage to current in the frequency domain. During analyte biorecognition and binding, the resistance and capacitance of an electric double-layer change cause variation in the impedance. By using small amplitude sine wave perturbation, linearity in electrochemical systems can be assumed, allowing the frequency analysis. EIS is further classified as a Faradaic or non-Faradaic EIS depending on the presence or absence of the redox indicator. The second one is more attractive due to no reagents being needed. Thus, the biorecognition process and label-free interactions on the sensor surface can be detected [36]. 

Recently, field effect transistors (FET), which are electronic semiconductor-based devices in which current flows are controlled by the applied electric field, started to populate the biosensor field. Depending on the type of the FET system, a p-type or n-type correlating with the type of their charge carriers, either positively or negatively charged analyte species can be sensitively detected by following the change in the conductance response. Novel nanotransducer designs (nanoparticles, nanotubes, nanowires, etc.) combined with biomolecule modifications demonstrate impressive sensitivity results, though the stability and sometimes selectivity of the bio-FET sensors may still be an issue [43].

## 3. Electrochemical Aptasensors

All biosensors can be classified according to the type of a signal transmitted, and are divided into electrochemical, acoustic, optical, and thermal/calorimetric biosensors [44,45]. Thermal biosensors measure the changes in temperature in the reaction between, for example, an enzyme and its analyte substrate. This change in temperature is then correlated with the amount of reactants consumed or products formed [46]. Resonant and acoustic wave biosensors operate by analyzing such measurand as a modulation in the physical properties of the acoustic wave that can then be correlated with the amount of the adsorbed analyte. These devices can also be miniaturised to get advantages in terms of size, scalability, and cost, and be easily integrated with microfluidics and electronics for multiplex detection across arrays of several devices implemented in a single chip [47]. Despite numerous reports, they have not yet found applications in clinical practice.

Optical detection biosensors are the most diverse class of biosensors exploiting many different spectroscopy techniques, such as UV-vis absorption, phosphorescence, luminescence or fluorescence. Of those, the most established commercial optical approach for protein analysis is the enzyme-linked immunosorbent assay: ELISA [48,49]. ELISA kits for specific protein cancer biomarkers are validated and widely used in clinical diagnostics of cancer; however, in some cases, they do not provide the sought sensitivity or specificity of analysis, which stimulates the search of advanced biosensor approaches [50]. Another popular type of optical biosensors is based on surface plasmon resonance and use the evanescent-wave phenomenon to describe interactions between receptors immobilised on the biosensor surface and ligands in solution. Binding of the analyte proteins by the surface-immobilised receptors alters the refractive index of the medium near the surface and this change is measured in real time to accurately estimate the amount of bound analyte, its affinity for the receptor and the dissociation and association kinetics of the interaction [51]. Despite many commercial devices offered on the market, due to their cost and often insufficient sensitivity they are not yet used in clinical practice.

Electrochemical transducers are also frequently used in biosensor research for the detection of cancer biomarkers, due to the advantages of their easy production, cost-effectiveness, and user-friendliness [44,49]. Electrochemical biosensors evaluate the current or potential response resulting from either the analyte binding (capacitive changes) or an oxidation and reduction reaction at the electrode surface. Of those, electrochemiluminescence (ECL) combining the electrogenerated chemiluminescent signal amplification with an optical read out allows improving the sensitivities of immunoassays and is currently used for clinical antibody-based analysis of a number of important analytes, including tumor biomarkers. Commercially available ECL analysers, among which is Roche cobas^®^ 6000, are quite efficient with their 170 to 2170 test h^−1^ [52], however, they may not be suitable for direct point-of-care testing (POCT). Similarly to optical ELISAs, they need a complex equipment for assay performance. Nevertheless, electrochemical detection schemes allow easy miniaturisation of the devices and production of portable devices, and here we will consider only this type of transducers. 

Combined with the aptamers, sensitive, accurate, and inexpensive electroanalytical approaches allow rapid bioanalyte detection with lower costs, smaller equipment size, and lower power requirements; they are easily adaptable for POCT by minimally trained personal [32,53]. Simple and robust electroanalytical schemes for specific and sensitive analysis of physiological-fluid-circulating protein biomarkers of cancer can support both prognosis of cancers and continuous monitoring of individual responses to cancer treatment therapies [54]. However, despite a large number of publications on the aptamer-based sensors, the majority of them are focused on just a few aptamer applications (almost 60% of all publications deal with only eight aptamers) [22,55], while the universal strategies for the aptamer applications in electrochemical bioassays are scarce. In the case of antibodies, the universal protocols have been much better established nowadays [56]. With each new aptamer-protein ligand couple, there is a need of a systematic research of their binding behaviour, conditions and structure of complexes formed, and, not the least, their interfacial behaviour [22].

In this context, the appropriate strategy for the aptamer immobilisation on the electrodes and other assay-related surfaces is crucial for the successful biosensor development. The choice of the immobilisation protocol strongly depends both on the physicochemical properties of the electrode surface and on the aptamer modifications. The ideal electrode material should enable the surface attachment of the aptamer in a controllable way, ensuring a strong and stable binding of the aptamer to the surface altogether with its conformational flexibility necessary for specific biorecognition of the targeted analyte [57]. The immobilisation protocol should also minimise a non-specific adsorption of matrix components and ensure the sensitive detection of the analyte. One of the important issues to be addressed in the protein aptasensor design is electrode surface fouling by blood serum and other biological fluid proteins: the non-specific interfacial adsorption often produces strong false positive signals even in the absence of the targeted proteins [58].

Aptamers can be assembled onto solid surfaces by several approaches that are very similar to those hitherto applied for immobilisation of single- or double-stranded DNA molecules [59,60,61]. Most common are physical adsorption, chemisorption, covalent attachment, and affinity interactions, such as avidin–biotin binding (Figure 3). Physical absorption of nucleic acid aptamers promoted by electrostatic interactions is the simplest strategy, which simultaneously results in the unsatisfactory stability of the electrostatically immobilised aptamers due to the relatively fast aptamer desorption [62] unless additional stabilisation of binding, e.g., to gold electrodes via thiol linkers, is used [63]. Aptamer self-assembly through chemisorption of the thiol linker-modified aptamers onto the gold surface is possibly the most popular approach, despite its suffering from insufficient long-term stability [64]. Typically, the alkanethiol-linker functionalised aptamers are self-assembled on gold electrodes through the sulphur-gold linkage complemented by filling the unblocked spaces/pinholes in the self-assembled monolayers (SAM) formed by the aptamers with alkanethiols [65,66], dithiols [67,68] or antifouling agents [69,70]. 

The most stable immobilisation is achieved through the affinity binding between the biotinylated aptamer sequence and avidin- [71], streptavidin- [72] or neutravidin-modified electrodes [73]. Each streptavidin and neutravidin molecule, possessing four biotin-binding sites, can bind up to two biotinylated aptamers (not all binding sites are accessible for reaction due to steric restrictions induced by these proteins adsorption on electrodes), which increases the number of aptamers on the sensor surface. It also reduces the non-specific protein adsorption and improves the sensor’s signal-to-noise (S/N) ratio [74]. For covalent bonds formation, the aptamers are correspondingly modified to react with the electrode surface functionalities [75].

Chemisorption, covalent binding, as well as chemical affinity interactions are most promising approaches of aptamers deposition due to the possibility of the aptamer one-point attachment to the electrode surface. However, the process of creating a covalent bond is quite complex and requires modification with appropriate functional groups of the electrode surface, aptamer sequence, or both. Table 1 summarises and compares the methods of aptamer deposition on solid surfaces [76,77].

The overall design of the biorecognition interface is always aimed at improving the sensitivity and the selectivity of the constructed aptasensor by a variety of signal amplification methodologies including regulating dimensionality, atomic arrangement and appropriate compositions of a sensing layer [78]. Electrocatalytic signal amplification approaches exploit enzymatic catalysis, electrocatalysis, and functional nanomaterials that can effectively enhance the aptasensor response by improving the interfacial conductivity and simultaneously diminishing the S/N ratio [79]. The latter usually requires a multi-step surface modification. Such nanomaterials as silica and noble metal (Au, Ag, Pt, Pd) nanoparticles (NPs), graphene oxide (GO), and carbon nanotubes and their nanocomposites and nanohybrids, polymers and metal (Zn, Zr, Ce, Hf, Gd, Sn, Mn, Fe) oxides are actively used in the electrochemical aptasensor construction [80,81,82,83,84,85,86,87,88,89]. Electrochemical nanomaterial-based aptasensors are numerously reported in biomedical research [90,91,92,93] and may satisfy a huge demand for portable analytical devices with the selectivity and specificity sufficient for healthcare applications, such as POCT of biomarkers for chronic and emerging diseases: cancer, neurodegenerative disorders, cardiovascular diseases and chronic respiratory infections [94]. The unique properties of the aptamers stimulate the further development of innovative principles of such electrochemical aptasensor operation [95,96], and currently the electrochemical aptasensor-related articles represent ca. 28% of the total number of publications on electrochemical biosensors [94]. However, despite the huge progress, electrochemical aptasensors have not yet entered the market [97].

Below, we discuss the selected examples of electrochemical aptasensors for the most important protein biomarkers of cancer and their suitability for biomedical assays. 

## 4. Electrochemical Assays for Cancer Biomarkers

### 4.1. Cancer Biomarkers

The National Cancer Institute (NCI) defines “biomarker” as “*a biological molecule found in blood, other body fluids, or tissues that is a sign of a normal or abnormal process, or of a condition or disease. A biomarker may be used to see how well the body responds to a treatment for a disease or condition. Also called molecular marker and signature molecule*” [98]. Cancer tumour biomarkers are of uttermost practical value in cancer screening, diagnosis, and evaluation of the effectiveness of anti-cancer therapies. They indicate the presence of malignancy or provide information about the likely future behaviour of cancer (i.e., the likeliness of progression or a response to therapy). In asymptomatic patients, tumour biomarkers can be used in screening tests for the early detection of malignant tumours. In symptomatic patients, biomarkers may help in the differential diagnosis of benign and malignant neoplastic lesions. After diagnosis and surgical removal of the neoplasm, analysis of biomarkers can allow assessing prognosis, postoperative observation, treatment prediction, and monitoring the response to systemic therapy [99].

Emerging and existing electrochemical aptaassays for tumour biomarkers are becoming an indispensable tool in precision medicine research. Due to their ultra-sensitivity, high selectivity, fast signal reading and simplicity, the electrochemical aptasensors may be the most suitable candidates for cancer theranostics. They are also ideally suited for POCT since they are inexpensive and easy to miniaturise and mass-produce. The following sections concentrate on the most representative examples of the developed electrochemical aptasensors for specific protein biomarkers of cancer: Human Epidermal growth factor Receptor-2; Urokinase Plasminogen Activator; Osteopontin; Mucin 1; Carcinoma antigen 125; Vascular Endothelial growth factor; Prostate-specific antigen; Platelet-derived growth factors; α-Fetoprotein and Carcinoembryonic antigen.

### 4.2. Human Epidermal Growth Factor Receptor-2

Human Epidermal growth factor Receptor-2 (HER-2/*neu*) is a 185 kDa glycoprotein complex associated with the receptor tyrosine kinase family. HER-2/*neu*’s overexpression occurs in several aggressive types of breast, oesophagus, and lung cancers that are characterised by the particularly aggressive growth and spreading of tumours [100]. Poor prognosis of these tumours and the necessity of their targeted therapeutical treatment require a continuous monitoring of the HER-2/*neu* state in response to the anticancer cure. All clinical methods for HER-2/*neu* detection rely on solid tumour biopsies and their PCR-based and FISH assaying [101,102] and are poorly suited for continuous HER-2/*neu* monitoring. 

Liquid biopsy analysis of HER-2/*neu* in cancer patients’ samples may be a possible alternative to solid tumour analysis: HER-2/*neu* released into the bloodstream by tumour cells has a huge biomarker potential to account for tumour heterogeneity compared to tissue biopsies. Several ELISA kits exploiting a sandwich assay construction exist on the market, for research purposes only, whose sensitivity and selectivity may be insufficient for precise determination of HER-2/*neu* at clinically requested, from 10^−11^ to 10^−10^ M HER-2/*neu*, serum levels (with a cancer cut-off at 2 × 10^−10^ M) [102]. 

A number of electrochemical liquid biopsy approaches were reported for sensitive and specific analysis of HER-2/*neu* in 0.5–4% serum samples that fitted the clinically requested concentration range [103,104,105]. An electrochemical immunoassay with [Fe(CN)_6_]^3−/4−^ as a redox indicator allowed as low as 0.01 ng mL^−1^ (1.4 × 10^−13^ M) HER-2/*neu* detection by differential pulse voltammetry (DPV) in 5% serum in 35 min [104] (NB: g mL^−1^ concentrations were recalculated in molar concentrations by taking into account 70 kDa MW of recombinant HER-2/*neu* used in the assays development). Here, to ensure electrode’s fouling resistance, antibody- and PEG-modified iron oxide nanoparticles were chemically linked to mercaptopropionic acid-modified gold nanoparticles (AuNP) electrodeposited on a gold electrode; the electrode surface was additionally blocked by bovine serum albumin (BSA). Sufficiently cumbersome bioconjugation and chemical modification protocols were used in the assay development, however. By using the aptamer as a biorecognition element, from 10^−12^ to 10^−8^ M HER-2/*neu* could be detected in a 30 min assay in 1% serum at the aptamer-modified Au electrodes via the electrocatalytically amplified ferricyanide reduction [103]. In a biosensor design, the HER-2/*neu*-specific thiolated aptamer and thiolated polyethylene glycol (PEG) were co-immobilised on gold; PEG prevented both the non-specific adsorption of serum albumins and the direct discharge of ferricyanide on the electrode surface, since electrochemistry of ferricyanide is inhibited by PEG (Figure 4A). Analysis of HER-2/*neu* binding to the aptamer in the presence of ferricyanide and methylene blue (MB), differently bound to the aptamer and the aptamer-HER-2/neu complex, allowed the enhancement of the electrocatalytic reduction of ferricyanide electrocatalysed by MB largely electrostatically bound to the protein–aptamer complex (Figure 4A). The assay was fast and simple, though it needed fast scan voltammetry for robust analysis of HER-2/*neu*.

Sandwich aptamer- and immunoassays with a catalytic signal amplification can further improve both the limit of detection (LOD) and specificity of HER-2/*neu* analysis. A nanoparticle-based hybrid sandwich immunoassay with silver enhancement allowed 0.1 pg mL^−1^ (1.4 × 10^−15^ M) HER-2/*neu* detection in 4% serum in 70 min [105]. The protein was trapped between the antibody immobilised onto the AuNP-glassy carbon electrode (GCE) and the aptamer reporter bearing hydrazine-modified AuNPs tags (Figure 4B). The signal amplification was provided by the silver-enhancement of the reporter AuNPs followed by square wave stripping voltammetry of the metal silver formed. 

A more traditional sandwich immunoassay on magnetic beads (MBs), with the alkaline phosphatase label and 1-naphtyl-phosphate substrate, detected 6 ng mL^−1^ (8.5 × 10^−11^ M) HER-2/*neu* in diluted serum in 2 h; PSA and cancer antigen 125 did not interfere [106]. 

The most promising performance was demonstrated by an electrochemical cellulase-linked sandwich ELISA on MBs [107]. Here, HER-2/*neu* was sandwiched between either antibody- or aptamer-modified MBs and antibody/aptamer reporters labelled with a non-redox active hydrolase - cellulase - an inexpensive and highly stable enzymatic label (Figure 4C). Applied onto graphite electrodes covered with an ultrathin insulating nitrocellulose film, the cellulase-labelled sandwiches digested the film. That resulted in the prominent changes in electrical properties of the electrodes, which was chronocoulometrically (CC) detected without any indicators present in solution (electrochemically label-free). Down to 1 fM HER-2/*neu* was detected in whole human serum samples in <3 h, with no interference from serum albumins or other cancer-related proteins, such as urokinase plasminogen activator protein. The best assay performance was achieved with the antibody-antibody-MBs and aptamer-antibody-MBs sandwich constructions (Figure 4C, right panel).

### 4.3. Urokinase Plasminogen Activator

Quite recently, urokinase plasminogen activator (uPA), a 54 kDa serine protease playing an important role in the urokinase activation system involved in cancer invasion and metastasis [108], was proposed as a universal prognostic biomarker of several cancer types [109,110]. Increased levels of uPA can be followed in ovarian and breast cancer, and squamous cell carcinoma, which makes uPA a valuable biomarker for liquid biopsy diagnosis of cancers [111,112]. The cancer cut-off value for this biomarker is 1.55 × 10^−11^ M (0.84 ng mL^−1^) [113].

A 33-mer RNA sequence specific for uPA was used for construction of the aptamer-modified electrodes for nM-pM analysis of uPA [114]. The RNA aptamer was stabilised against the ribonuclease digestion by its selection from the pool of fluorinated RNAs, in which fluorine substituted hydrogen in the 2’ hydroxyl group of the ribose ring. Chemical modifications of fluorinated aptamers are expensive, and for the aptamer attachment to the Au electrode a phosphorothioated adenosine dA* tag was introduced enzymatically into the 3’-end of the aptamer sequence. Nucleic acid (NA) immobilisations onto gold via the dA* tags showed improved binding stability compared to the regular alkanethiol linkage [115] and cheapened the fluorinated RNA aptamer modification compared to the automated NA synthesis. After the aptamer tethering to gold through the dA* linker, the surface was blocked with mercaptohexanol. The uPA binding was interrogated in two ways: at negative electrode potentials, with MB as a redox indicator of uPA-aptamer binding, and at positive potentials, with ferricyanide as a solution-diffusing redox indicator sensitive to the electrode surface blocking by the formed uPA-aptamer complex. A non-specific adsorption of BSA interfered with the uPA detection at positive potentials, which resulted in 1 nM LOD insufficient for liquid biopsy analysis. With MB, a much more specific 1 pM analysis of uPA allowed its robust detection in serum, which resulted both from the different mechanism of the redox indicator-uPA-aptamer interactions and minimised interference from serum components [116]. The electrochemical modulation of the aptamer surface state was shown to be the key factor in optimisation of this electrochemical aptasensor response to uPA. 

### 4.4. Osteopontin

Osteopontin (OPN), also known as a transformation-related protein phosphatase, is an extracellular matrix-secreted phosphorylated glycoprotein. It plays a major role in such physiological processes as bone remodelling, inflammations, immune-regulation and vascularisation. OPN is a major mediator of inflammation—a key factor in carcinogenesis with multi-functional activities [117,118]. The up-regulation of OPN expression has been identified in a variety of human cancers, including but not limited to breast [119], ovarian [120], prostate [121], and oral cavity cancers [122], lung [123], liver [124], gastric [125], pancreatic [126], and colorectal cancers [127], glioma [128], thyroid carcinoma [129], and melanoma cancer [130]. The WHO technical report identifies breast cancer as the most common cancer among women, with the highest incidence in 2018 (11.6% of all cancers) [1]. Rapid detection of its potential biomarker—human OPN—offers a great promise for its rapid POCT (cancer cut-off at 8.54 × 10^−12^ M) [131].

Several electrochemical aptasensors for detection of OPN have been reported. Cao and co-workers performed the OPN detection at a cucurbit[7]uril (CB[7])-functionalised pyrolytic graphite electrode. Binding of OPN to its aptamer in the test solution was followed by an exonuclease-catalysed digestion of MB-labelled DNA oligonucleotides [132]. The CB[7] molecules immobilised on the electrode captured the released MB-labelled nucleotides that accumulated on the electrode surface and subsequently yielded the voltammetric response related to the concentration of OPN. This assay combined the host–guest properties of CB[7] with the immobilisation-free homogeneous assay, which provided a linear response to OPN in the range from 50 to 500 ng mL^−1^ (from 7.9 × 10^−10^ M to 7.9 × 10^−9^ M) with LOD of 10.7 ng mL^−1^ (1.7 × 10^−10^ M). OPN was assayed in 10-fold-diluted human serum samples spiked with the protein [132].

A simpler aptasensor for OPN exploited ferri/ferrocyanide as a redox indicator and the affinity immobilisation of the aptamer [133]. The gold electrode was modified with 3,3-dithiodipropionic acid and its carboxylic groups were activated by EDC/NHS coupling chemistry. Then, a streptavidin layer was formed by the covalent attachment to the activated COOH groups, further reacting with a biotinylated 40-mer RNA aptamer. The protein binding blocked the sensor surface, and the voltammetric response from ferricyanide decreased with the increasing concentration of OPN from 25 to 2402 nM, with the signal saturation observed at 800 nM OPN. The aptasensor showed LOD of 3.7 nM OPN within the range reported for patients with metastatic breast cancer. This aptasensor could detect OPN in the presence of lysozyme, bovine osteopontin and BSA, however, thrombin interfered [133]. Later, the same group improved this assay by using a new biotinylated DNA aptamer [134]. The DNA-based aptasensor showed better LOD of 2.6 nM OPN (detection by cyclic voltammetry, CV) and 1.4 nM OPN (by square wave voltammetry, SWV) in buffer solutions. LOD of 1.3 nM OPN detected by SWV in OPN-spiked synthetic human plasma was within the OPN plasma levels reported for patients with breast cancer (0.4–4.5 nM) and recurrent/metastatic breast cancer (0.9–8.4 nM). The results in human plasma were comparable with those obtained by ELISA. Thrombin generated current signals 2.6–10 times lower than OPN [134].

Zhou and co-workers proposed another strategy for the OPN detection based on the ferricyanide redox indicator and use of nanocomposite materials both for the aptamer immobilisation and biofouling prevention (Figure 5) [135]. The aptamer was immobilised on the nanohybrid of Ti_3_C_2_T_x_ MXene and phosphomolybdic acid (PMo_12_) embedded within polypyrrole (PPy@Ti_3_C_2_Tx/PMo_12_). PPy@Ti_3_C_2_Tx/PMo_12_ showed good stability and biocompatibility, and enabled a strong binding of the aptamer. Down to 0.98 fg mL^−1^ OPN was detected by electrochemical impedance spectroscopy (EIS) in the presence of [Fe(CN)_6_]^3−/4−^. This aptasensor was tested in spiked human serum samples and showed low interference from thrombin, BSA, immunoglobulin G (IgG), immunoglobulin E (IgE), lysine, and prostate-specific antigen [135]. Another nanocomposite material for the aptamer anchoring, a hybrid of zirconium oxide nanoparticles and graphene-like nanofiber (ZrO_2_@GNF), did not improve LOD (then 4.76 fg mL^−1^ or 7.5 × 10^−17^ M), but allowed from 0.01 pg mL^−1^ to 2.0 ng mL^−1^ (from 1.6 × 10^−16^ M to 3.2 × 10^−11^ M) OPN detection in spiked human plasma samples, with a negligible interference from some other cancer biomarker proteins and IgG [136].

### 4.5. Mucin 1

Mucin 1 (MUC1, CA-15-3) belongs to the group of transmembrane glycoproteins. It is aberrantly glycosylated and overexpressed in bladder, breast, colon, lung, prostate, pancreatic, and ovarian carcinomas and is a useful biomarker for diagnosis of early cancers and evaluation of tumour-related diseases [137,138,139]. Being highly overexpressed in breast cancer, human MUC1 is one of the most common tumour biomarkers for this disease diagnosis [137] (the cancer cut-off level of MUC1 is 3.96 × 10^−12^ M [140]). Unsurprisingly, a large number of electrochemical aptasensors for MUC1 detection has been suggested, and some selected examples are summarised in Table 2.

One of the pioneer electrochemical aptamer assays for MUC1 is an electrochemical sandwich assay with a nanocomposite label [141]. It combined a dual signal amplification strategy of the poly(o-phenylenediamine)–AuNPs (PoPD–AuNPs) hybrid film used as a support for the aptamer immobilisation and the thionine-AuNPs-functionalised silica/multiwalled carbon nanotubes (MWCNT) core–shell nanocomposite (Thi-AuNPs/SiO_2_@MWCNTs) as a tracing label. PoPD was electropolymerised on the gold electrode surface and modified further with AuNPs and the thiolated aptamer, mercaptohexanol being used as a backfiller. After reaction with MUC1, the AuNPs–PoPD/aptamer/MUC1-modified electrodes were exposed to the aptamer/Thi-AuNPs/SiO_2_@MWCNTs nanoprobes to form the aptamer sandwich complex with MUC1. MUC1 was detected through the DPV response from thionine: the DPV peak currents gradually increased with MUC1 concentrations increasing from 0 to 10^−7^ M MUC1. Under the optimised conditions, the aptasensor showed the linear dynamic range from 1 to 100 nM and LOD of 1 pM. Carbohydrate antigens 19-9 and 72-4, carcinoembryonic antigen and BSA did not interfere with the MUC1 detection [141].

Liu and co-workers proposed another aptasensor solution based on AuNPs-related signal amplification [142]. A thiolated capture DNA probe (SH-cDNA), partially complementary to the aptamer sequence, was co-immobilised with mercaptohexanol (MCH) on the gold electrode. Next, the SH-cDNA/MCH-gold surface was modified via hybridisation in two ways: (I) directly with the aptamer or (II) with the signal enhancer, the aptamer-AuNPs conjugate (Apt@AuNPs). Binding of MUC1 induced the interfacial changes and, as a result, changed the EIS response from [Fe(CN)_6_]^3−/4−^. The sensitivity of the MUC1 detection was improved with Apt@AuNPs conjugates, with LOD of 0.1 nM, and human blood serum samples were analysed by the standard addition method [142].

In another nanocomposite design, a competitive electrochemical aptasensor exploited a cDNA-ferrocene/MXene complex, in which MXene (Ti_3_C_2_) nanosheets were nanocarriers for the complementary DNA-ferrocene probe (cDNA-Fc, partially complementary to the MUC1 aptamer sequence) [143]. The sensor preparation involved three steps: (1) binding of the cDNA-Fc probe to MXene, (2) modification of AuNPs-modified GCE with the thiolated aptamer, and (3) competitive recognition of MUC1. The cDNA-Fc/MXene probe was coupled to the aptamer/AuNPs/GCE. The resulting cDNA-Fc/MXene/Apt/Au/GCE aptasensor was used for the MUC1 detection. The competitive binding reaction between the cDNA-Fc/MXene probe and MUC1 caused the release of the cDNA-Fc/MXene probe from the sensor surface. A linear relation between the detected SWV currents and the MUC1 concentration was followed for 1.0 pM to 10 mM MUC1, with LOD of 0.33 pM. The aptasensor detected MUC1 in human serum samples analysed by the standard addition method [143].

An even simpler in construction electrochemical aptasensor was designed using screen-printed carbon electrodes (SPCEs) modified with MWCNTs functionalised by using diazonium salt chemistry for the aptamer immobilisation and a ferricyanide redox indicator [144]. The grafted benzoic acid functional groups were further activated by EDC and reacted with the amino-terminated aptamer; in addition, the electrode was blocked by BSA to deactivate the remaining terminal groups and unreacted sites. The aptamer-MUC1 binding followed by EIS in the presence of [Fe(CN)_6_]^3–/4–^ allowed down to 0.02 U mL^−1^ MUC1 detection with a linear range from 0.1 to 2.0 U mL^−1^. The aptasensor operated well in human blood serum, with negligible interferences from BSA, lysozyme, and fetal bovine serum (FBS) [90]. Immobilisation of the aptamer on the AuNPs and GO-doped poly(3,4-ethylenedioxythiophene) (PEDOT) nanocomposite films electropolymerised on the surface of fluorine tin oxide (FTO) electrodes allowed further improvement of MUC1 analysis [145]. The biotinylated aptamer was immobilised via biotin-avidin linkage, and LOD of 1 fg mL^−1^ (0.031 fM) was shown by DPV with [Fe(CN)_6_]^3−/4−^. The fabricated device could determine MUC1 in spiked human serum samples with 85–93% recovery [145].

The MUC1 was also explored as a biomarker in the electronic beacon-based prostate cancer assay [146]. A thiolated DNA hairpin aptamer was conjugated to the MB redox label and immobilised onto the gold electrode via the alkanethiol linker, MCH being used as a co-adsorbant to backfill the pinholes and remove non-specifically adsorbed aptamer molecules. The aptasensor detected down to 0.65 ng mL^−1^ (15.9 pM) MUC1 by SWV, within the 0.65–110 ng mL^−1^ (5.3 × 10^−12^–9.0 × 10^−10^ M) concentration range. The MUC1 production patterns were precisely determined in benign (RWPE-1) and prostate cancer cells (LNCaP and PC3), with no significant interference from ascorbic and uric acids, vascular endothelial growth factor, BSA, and prostate specific antigen [146].

**Table 2 sensors-21-00736-t002:** Analytical performance of selected electrochemical aptasensors for MUC1 detection.

Strategy	Technique	LOD, M	Interference Studies	Ref.
POPhDA–AuNPs hybrid film/AuNPs silica/MWCNT C-Sh/Au	DPV	1 × 10^−12^	CEA, CA19-9, CA72-4, BSA	[141]
ALP-strp/Apt2/Biotin/Apt1/Strp-MBs/SPA		7 × 10^−11^	MUC4, MUC16	[146]
Apt-AuNPs/MCH/cDNA/Au	EIS	10^−10^	TNF-α, CEA	[142]
Exo I/MUC1/Apt-MB/CP/MCH/Au	SWV	4 × 10^−12^	Myo, BSA, CEA	[147]
Apt/EDC/MWCNT/SPCE	EIS	0.02 U mL^−1^	BSA, FBS, Lyz	[90]
Th/rGO-N^′1^,N^′3^ DHMIA/Apt/MUC1/Apt/Pdots/IL/Au	DPV	6 × 10^−11^	MUC4, Lys, Myo	[148]
SH-Apt-MB/Au	SWV	4 × 10^−9^	AA, UA, VEGF, BSA, PSA	[145]
MB-Apt/AuNPs/GCE	DPV, EIS	24 × 10^−9^	Lyz, BSA, Cyt C	[149]
Apt/ZrHCF NPs/ZrHCF/mFe_3_O_4_/mC/Au	EIS	7.4 × 10^−15^ (0.9 pg mL^−1^)	CEA, IgG, BSA	[150]
AuNPs and GO doped PEDOT films APT/Strp/AuNPs-GO-PEDOT	DPV	3.1 × 10^−17^	MPT64, AChE, BSA	[144]
Exo I/Apt-MUC1/cDNA-MB/Naf/ITO		3.3 × 10^−15^ M (0.4 pg mL^−1^)	CEA, GP73, HSA, ALP, AFP	[151]
Metal ion electrochemical labels/Ru(NH_3_)_6_^3+^ electronic wires		3.33 × 10^−15^	FBS, HCG, MUC16, CA19-9	[152]
AuNPs-DNA enzyme/H-2/MCH/c-DNA/Au	Amp	3.3 × 10^−16^ (0.04 pg mL^−1^)	PSA, Thr, CEA, BSA	[153]
MXene probe/c-DNA-Fc/Apt + BSA/AuNPs/GCE	SWV	3.3 × 10^−13^	Not shown	[143]

Abbreviations: POPhDA—poly(o-phenylenediamine); AuNPs—gold nanoparticles; MWCNs—multi-walled carbon nanotubes; C-Sh—core-shell; Au—gold electrode; ALP—alkaline phosphatase; Strp—streptavidin; Apt; Apt1; Apt2—aptamers; MBs—magnetic beads; SPA—screen-printed arrays; MCH—6-mercaptohexan-1-ol; cDNA—capture DNA; Exo I—Exonuclease I; MB—methylene blue; MUC1; MUC4; MUC16—transmembrane mucins; SWV—Square Wave Voltammetry; EIS—electrochemical impedance spectroscopy; GCE—glassy carbon electrode; DPV—Differential Pulse Voltammetry; Amp—amperometry; CEA—carcinoembryonic antigen; CA19-9—carbohydrate antigen 19-9; CA72-4—cancer antigen 72-4; BSA—bovine serum albumin; TNF-α—tumour necrosis factor α; Myo—myoglobin; Lyz—lysozyme; FBS—fetal bovine serum; Lys—lysine; AA—ascorbic acid; UA—uric acid; VEGF—vascular endothelial growth factor; PSA—prostate specific antigen; Cyt C—cytochrome C; IgG—Immunoglobulin G; MPT64—Mycobacterium tuberculosis; AChE—acetylcholinesterase; GP73—golgi protein 73; HSA—human serum albumin; AFP—alpha-fetoprotein; HCG—human chorionic gonadotrophin; Thr—thrombin; EDC—1-Ethyl-3-(3-dimethylaminopropyl) carbodiimide; SPCE—screen-printed carbon electrode; IL—interleukin; Pdots—polymer dots; rGO—reduced graphene oxide; DHMIA—N^′1^,N^′3^ dihydroxymalonimidamide; Th-thionine; ZrHCF NPs—zirconium hexacyanoferrate nanoparticles; mFe_3_O_4_—mesoporous mFe_3_O_4_; GO—graphene oxide; PEDOT—poly(3,4-ethylenedioxythiophene); Naf—Nafion; ITO—indium tin oxide electrode; H-2—hairpin 2; Fc—ferrocene; MXene—two-dimensional inorganic compounds.

### 4.6. Carcinoma Antigen 125

Carbohydrate antigen 125 (CA125), also known as mucin 16 (MUC16), is a heavily O-glycosylated protein and a component of the female reproductive tract epithelia, the respiratory tract and the ocular surface. It is aberrantly overexpressed in breast, ovarian, lung and pancreatic tumours, and thyroid cancers and plays an important role in cancer progression and metastasis [154,155,156]. The world’s incidence of gynaecological tumours is high, with ovarian cancer being one of the most common malignancies in the female reproductive system with the highest mortality rate. In the early stage of cancer development, most patients do not show any clinical signs or symptoms, and since its discovery in 1981, CA125 has been used as a gold standard biomarker of epithelial ovarian cancer [157]. The concentration of CA125 in ovarian cancer patients’ samples ranges between 5.4 to 6700 U mL^−1^, with the threshold level (the cut-off value above which the result is considered positive) at 35 U mL^−1^ [158].

Both the high incidence and mortality rate of ovarian cancer, particularly among the European female population, require robust and inexpensive liquid biopsy tests to timely diagnose this cancer, and many efforts are focused on the development of appropriate electrochemical aptasensors for the CA-125 detection.

The field-effect aptasensor detected down to 5 × 10^−10^ U mL^−1^ CA125, also in serum samples from ovarian cancer patients [159]. This flexible field-effect-transistor (FET) aptasensor was constructed by non-covalent immobilisation of carboxylated MWCNTs onto a few layers of reduced GO (rGO) nanosheets integrated with a poly-methyl methacrylate substrate; the aptamer was attached to carboxylated MWCNTs via EDC/NHS chemistry. The electrical characteristics of the constructed liquid-ion-gated FET aptasensor were followed by measuring the FET currents resulting from the protein binding. Under optimal conditions, the aptasensor showed the linear dynamic range from 10^−9^ to 1 U mL^−1^ of CA125, with negligible interference from carcinoembryonic antigen, alpha-fetoprotein, and cancer antigen 15-3 [159].

The promising electrochemical aptasensing platform was elaborated by combining the aptamer modification of SPCEs with the target-triggered strand displacement strategy (Figure 6) [160]. The thiolated DNA hairpin sequence (SH-hDNA) was immobilised onto flower-like gold nanostructures electrodeposited onto polylayers of poly(ethylene imine)/poly(acrylic acid) (PEI/PAA) layer by layer self-assembled on SPCEs. The modified SPCE surface was blocked with MCH to remove non-specifically adsorbed DNA. Further, the hybrid was formed between the DNA probe partially complementary to SH-hDNA and the CA125 aptamer and exposed to CA125 samples. CA125 binding released the DNA probe that then hybridised with the toehold of SH-hDNA. The second hairpin labelled with MB (hDNA-MB) then subsequently hybridised via its toehold domain with SH-hDNA, and the increasing SWV response from the MB label generated after these cascade reactions was linearly proportional to the CA125 concentration within the 0.05 to 50 ng mL^−1^ range (5.0 × 10^−14^ to 5.0 × 10^−11^ M). LOD was 5 pg mL^−1^ (5.0 × 10^−15^ M), and a negligible interference from BSA, PSA, CD63 and epithelial cell adhesion molecule was observed. The aptasensor could operate in CA125-spiked serum, urine and saliva [160].

A nanocomposite electrode modification exploiting amidoxime-modified polyacrylonitrile electrospinned nanofibers (NFs) decorated with Ag nanoparticles (AgNPs-PAN-oxime NFs) allowed the 0.0042 U mL^−1^ DPV detection of CA125 and its clinical applications, such as analysis of CA125 positive and negative human serum samples comparable to ELISA [161]. In sensor construction, the indium tin oxide (ITO) electrode was covered by AgNPs-PAN NFs, chemically modified to bear amidoxime groups, to which the aminated aptamer was coupled via glutaraldehyde (GA) cross-linking. The modified electrode was blocked with BSA to prevent non-specific binding. The MB-labelled signalling-probe (sDNA) was further immobilised on the ITO/AgNPs-PAN NFs/NH_2_-aptamer surface through its partial hybridisation to the aptamer. The sensor was exposed to CA125, and its recognition by the aptamer released the MB-labelled sDNA into solution. The sDNA displacement resulted in the diminishing MB peak currents detected by DPV within the 0.01 to 350 U mL^−1^ CA125 dynamic linear range. Lysozyme, IgG, IgA, BSA and serum haemoglobin (Hb) did not significantly influence the CA125 analysis [161].

Metal–organic frameworks (MOFs), formed by coordination bonds between metal ions and organic ligands, were shown to be a suitable platform for the aptamer immobilisation due to their good adsorption capacities [162]. Two kinds of bimetallic core-shell Tb-MOF-on-Fe-MOF and Fe-MOF-on-Tb-MOF nanostructures were deposited on gold electrodes, and the aptamer was immobilised on those, the Tb-MOF-on-Fe-MOF composite demonstrating the best adsorption characteristics. CA125 binding followed by EIS with [Fe(CN)_6_]^3−/4−^ allowed down to 58 μU mL^−1^ CA125 analysis (a linear range from 0.1 mU mL^−1^ to 200 U mL^−1^). The aptasensor performed well in human serum, with no interference from MUC1, carbohydrate antigen 19-9, vascular endothelial growth factor, immunoglobulin G, carcinoembryonic antigen, epidermal growth factor receptor, porcine serum albumin, and alpha-fetoprotein [162].

A hybrid aptamer-antibody sandwich assay based on the hybridisation chain reaction (HCR), with MB as a redox indicator, could detect 0.02 U mL^−1^ CA125 in human samples [163]. The antibody/CA125/aptamer sandwich was formed by reaction of the CA125-specific antibody immobilised onto the surface of the AuNPs-modified ITO electrode with CA125, stepwise reacting with its aptamer. Then the 3′-end of the aptamer was subjected to HCR. DPV oxidation signals from the redox indicator intercalating into the G-C reach regions of the formed duplexes were linearly proportional to from 0.39 to 200 U mL^−1^ of CA125, and uric acid, alpha-fetoprotein, carcinoembryonic antigen and cancer antigen 153 did not interfere with the sensor response [163]. Another variant of the aptamer/CA125/antibody sandwich assay was performed on MBs [164]. COOH-functionalities of MBs were activated in reaction with EDC and then allowed to react with CA125 monoclonal antibodies and horseradish peroxidase (HRP) as an enzymatic label. The antibody-modified MBs were used to capture the CA125 from human blood and serum samples and then collected on the aptamer-modified gold electrodes via the CA125 sandwich aptacomplex formation. HRP-labelled MBs attached to electrodes generated the bioelectrocatalytic response in the presence of its substrate H_2_O_2_, and CV and EIS responses to CA125 were linear for 2 to 100 U mL^−1^ CA125, covering the entire clinical concentration range of serum samples [164].

### 4.7. Vascular Endothelial Growth Factor

Vascular endothelial growth factor (VEGF) is a growth factor and signalling protein involved in tumour angiogenesis by increasing the blood vessel permeability. It plays a crucial role in the endothelial cell growth, proliferation, migration, and differentiation. VEGF is considered as an important biomarker for such diseases as cancer, rheumatoid arthritis, retinopathy, and some neurodegenerative diseases, such as Parkinson’s and Alzheimer’s diseases. Aberrant expression of VEGF occurs in many types of cells, including cancer cells. Elevated levels of VEGF are often detected in patients with breast, lung, ovarian, thyroid, and prostate cancers [165,166,167].

An impressive number of electrochemical aptasensors for VEGF analysis has been reported during last five years. Table 3 summarises some representative examples focused on electroanalysis of human VEGF_165_—the recombinant form of a 38.2 kDa homodimeric glycoprotein composed of two identical 165 amino acid chains. The VEGF_165_ levels in healthy people range between 60.3 fM and 12.2 pM (2.30 and 467.10 pg mL^−1^) [168], with the VEGF_165_ cut-off value of 6.7 pM (or 256 pg mL^−1^) in cancer patients [169].

In 2018, a comprehensive review by Dehghani and co-workers summarised the research on the electrochemical aptasensors for VEGF_165_ [170], and here we provide an overview of the works appeared since then. Most of them exploit different types of redox probes, including the routine ferri/ferricyanide redox indicator, Fc and MB labels, and more complex redox-nanocomposite modifications.

The cucurbituril and azide co-functionalised graphene oxide CB[7]-N_3_-GO nanocomposite linked to Fc-modified branched ethylene imine polymer (BPEI) (BPEI-Fc/CB[7]-N_3_-GO) via host–guest interactions was suggested as an electrochemical label for the VEGF_165_ aptasensing [171]. The solution-formed aptamer/alkyne-DNA hybrid released the alkyne-DNA strand after reaction with VEGF_165_, and this alkyne-DNA reacted with the hairpin DNA immobilised on the gold electrode surface through the thiol linker. After the hybrid formation, the duplex was labelled through the “electro-click” reaction with BPEI-Fc/CB[7]-N_3_-GO producing the change in the electrochemical response within the 2.62 × 10^−16^ to 2.62 × 10^−11^ M (10 fg mL^−1^ to 1 ng mL^−1^) VEGF_165_ range and LOD of 2.1 × 10^−16^ M (8 fg mL^−1^) [171].

Another nanocomposite approach used the Au-Pd alloy/EMIMPF6 ionic liquid (IL) modification of GCE as a platform for the aminated aptamer immobilisation via the glutaraldehyde coupling reaction [145]. BSA co-immobilisation combatted the non-specific binding. The DPV response from the MB redox indicator (“signal off”) and the EIS response from [Fe(CN_6_)]^3−/4−^ (“signal on” assay) displayed the linear ranges of 1–150 pM and 5–200 pM, respectively, and LOD of 0.5 and 0.78 pM. The aptasensor detected VEGF_165_ as a lung cancer biomarker in human serum samples [172].

Gold/graphene quantum dot/thionine nanocomposite hybrid (Au/GQD) allowed even more pronounced amplification of the aptasensor response [173]. A thiolated hairpin DNA probe H2 (MCH as a co-adsorbant) was immobilised on gold electrodes, and further the surface was modified to produce a hybrid consisting of DNA hairpin H1-Au/GQD-thionine. Each VEGF_165_ molecule could bound to two other DNA probes via the specific aptamer-target recognition to obtain a molecular machine that reacted/hybridised with the surface-tethered H2 probe through the proximity effect. The aptasensor detected down to 0.3 fM VEGF_165_, also in human serum, while the SWV signal increased linearly within the 1 fM to 120 pM concentration range [173].

Another impedance aptasensor exploited the ferricyanide redox indicator and a nanocomposite consisting of gold nanoarchitectures (AuNAs) embedded with nanochitosan (NChtn) for the aptamer immobilisation. It allowed a 1.77 × 10^−13^ M (6.77 pg mL^−1^) VEGF_165_ detection. The AuNAs@NChtn-based aptasensor demonstrated reasonable suitability for the VEGF_165_ detection in real serum samples [174].

A ratiometric aptasensor exploited two sensing biointerfaces for VEGF detection: a GO/MB redox nanocomposite either covalently attached or physically absorbed on GCE with a Fc-labelled aptamer assembled on the top. The linear range of the designed GCE-GO/MB-streptavidin/biotin-aptamer-Fc sensors was 2.62 × 10^−13^–1.31 × 10^−11^ M and 5.24 × 10^−13^–1.31 × 10^−11^ M (10–500 pg mL^−1^ and 20–500 pg mL^−1^) VEGF for covalent and non-covalent attachment of GO/MB, respectively (Table 3) [175]. Another redox-nanocomposite aptasensor for real-time detection of VEGF in serum was based on the rGO/MB-AuNPs nanocomposite deposited on GCE, on which the mixture of SH-aptamer-Fc and the thiol-modified antifouling agent—polyethylene glycol (SH-PEG)—was self-assembled [176]. The VEGF binding decreased the intensity of both Fc and MB signals due to the electrode reactions impeded by the formed VEGF/aptamer-Fc complex. The ratiometric dual signal from Fc and MB allowed LOD of 2.62 × 10^−15^ M (0.1 pg mL^−1^) and the linear range of 5.24 × 10^−14^ to 1.31 × 10^−11^ M (2–500 pg mL^−1^) VEGF [176].

Enzymatic bioconjugate labels also allowed their easy integration with the aptamer sequences. Glucose dehydrogenase (GDH) conjugated with DNA-binding protein—zinc finger protein (ZFP)—was used to label the DNA aptamer specific for VEGF by taking advantage of the sequence-specific binding ability of ZFP [177]. For the construction of the hybrid sandwich immunosensor, streptavidin was covalently attached to dithiobis(succinimidyl undecanoate) SAMs formed on a gold wire and then used for immobilisation of the biotinylated anti-VEGF antibody. VEGF binding was amperometrically detected in the presence of glucose, at 105 pM VEGF levels, after VEGF labelling with the aptamer-GDH-ZFP complex [177].

An impedance line-pad-line electrode (LPLE) aptasensor allowed label-free 0.017 fM electroanalysis of VEGF_165_ [178]. Impedance responses of the aptamer-modified gold LPLE (BSA as a blocking agent) were linearly proportional to 0.026–31.4 fM VEGF_165_ with insignificant interference from such proteins as thrombin, PDGF-BB; VEGF_121_, and human IgG [178]. These results are remarkable considering the massive efforts focused on the development of the label- (and indicator-) free approaches. Another label-free sensor for VEGF analysis in human blood exploited the complementary metal oxide semiconductor (CMOS) platform [23]. The VEGF binding by the peptide-aptamer-modified microneedles was followed by monitoring the capacitance changes between the microneedles by a two-step capacitance-to-digital converter (CDC). The aptasensor detected VEGF at its 0.1 pM levels [23].

**Table 3 sensors-21-00736-t003:** Analytical performance of selected electrochemical aptasensors for VEGF_165_.

Strategy	Technique	LOD, M	Interference Studies	Ref.
Sandwich Apt/MBs-Ab/VEGF/Apt/AuIDE	EIS (capacitance)	10.5 × 10^−12^ (401 pg mL^−1^)	BSA	[179]
NP/Strp-ALP/B-Apt/VEGF/MCH/SH-Apt/AuNPs/SPGE	DPV	3 × 10^−8^	HER2	[180]
MB/Apt/GA/BSA-AuNCs/IL/GCE Apt/GA/BSA-AuNCs/IL/GCE	DPV EIS (Ferri/ferrocyanide)	signal-off: 0.32 × 10^−12^ signal-on: 0.48 × 10^−12^	Not shown	[169]
SH-Apt/OMC–Au_nano_/SPCE	EIS (Ferri/ferrocyanide)	26.2 × 10^−15^ (1 pg mL^−1^)	HIgG, HIgA, Lip, Lyz, HSA	[181]
Fc-Apt-alkyne/UDT + UDT-N_3_/Au	ACV	6.2 × 10^−9^	VEGF_121_, BSA, HSA, trypsin	[182]
DNA-Ag/Pt NCs/amino-Apt/GCE Peroxidase mimicking activity	Amp	4.6 × 10^−12^	Thr, HSA, HIgG	[183]
MCH/SH-Apt-MB/Au	SWV	3.93 × 10^−12^ (0.15 ng mL^−1^)	AA, UA	[184]
Apt/Au_nano_/rGO-PAMAM-Th/SPCE	DPV	0.7 × 10^−12^	U, AA, D, Glu, HIgG, HIgA	[185]
BPEI-Fc-CB[7]-N_3_-GO/S1/MCH/S2/SH-Apt/Au	SWV	0.21 × 10^−15^ (8 fg mL^−1^)	BSA, HSA, VEGFR1, VEGFR2, VEGF_121_	[171]
MB/NH_2_-Apt/GA/AP/IL/GCE NH_2_-Apt/GA/AP/IL/GCE	DPV EIS (Ferri/ferrocyanide)	signal-off: 0.5 × 10^−12^ signal-on: 0.78 × 10^−12^	Not shown	[172]
Hairpin DNA/AuNPs/GQD/Thi/MCH/SH-hairpin DNA/Au	SWV	0.3 × 10^−15^	PSA, BSA, Thr	[173]
Peptide Apt-based functionalised microneedles	EIS (capacitance)	0.1 × 10^−12^	HIgG, Con A, cholera toxin	[23]
SH-Apt/AuNAs@NC/Au	EIS (Ferri/ferrocyanide)	0.18 × 10^−12^ (6.77 pg mL^−1^)	Lyz, HIgG, CEA, PSA	[174]
SH-Apt/LPLE	EIS	0.017 × 10^−15^ (0.64 fg mL^−1^)	Thr, PDGF-BB; VEGF_121_, HIgG	[178]
Sandwich Apt-GDH/VEGF/Biotin-Ab/ strp/DTBSU/AuWE	Amp	105 × 10^−12^	BSA	[177]
biotin-Apt-Fc/MB-strp/GO-GCE biotin-Apt-Fc/MB-strp/GO-PhA-GCE	SWV	2.62 × 10^−15^ (1 pg mL^−1^) 0.18 × 10^−12^ (7 pg mL^−1^)	IL-6, CA-125, PSA, HIgG	[175]
SH-Apt-Fc + SH-PEG/AuNPs-MB/rGO-GCE	SWV	2.62 × 10^−15^ (0.1 pg mL^−1^)	HIgG, IL-1β, PSA, IL-6	[176]

Abbreviations: AuIDE—interdigitated electrodes; Apt-aptamer; MBs—magnetic beads; Ab—antibody; VEGF—vascular endothelial growth factor; SWV—Square Wave Voltammetry; EIS—electrochemical impedance spectroscopy; DPV—Differential Pulse; Amp—amperometry; NP—1-naphthyl-phosphate; Strp-ALP—streptavidin-alkaline phosphatase conjugate; B-Apt—Biotinylated Apt; AuNPs/SPGE—Gold nanostructured graphite screen printed electrodes; GA—glutaraldehyde; BSA-AuNCs—BSA-gold nanoclusters; IL—ionic liquid; GCE—glassy carbon electrode; Au/GQD—Gold/graphene quantum dot hybrid; OMC–Au_nano_ mesoporous carbon–gold nanocomposite; HER2—human epidermal growth factor receptor 2; HIgG—human immunoglobulin G; HIgA—human immunoglobulin A; Lip—lipase; Lyz—lysozyme; HSA—human serum albumin; UDT-N_3_—11-Azido-1-undencanethiol; ACV—alternating current voltammetry; DNA-Ag/Pt NCs—DNA bimetallic Ag/Pt nanoclusters; Thr—human thrombin; SPCE—screen-printed carbon electrode; rGO-PAMAM/Au_nano_–reduced graphene oxide/gold functionalised with poly(amidoamine) dendrimers; U—urea; AA—ascorbic acid; D—dopamine; Glu—glucose; BPEI–branched ethylene imine polymer; Fc—aminomethylferrocene; CB[7]-cucurbit[7]urils macrocycle; N3-GO–azide-functionalised graphene oxide; S1—alkyne-functionalised DNA strand; MCH—6-mercaptohexan-1-ol; S2–hairpin DNA strand; AP–Au-Pd alloy; IL–ionic liquid; Con A—concanavalin A; AuNAs@NC–Au nanoarchitecture embedded with nanochitosan; CEA—carcinoembryonic antigen; LPLE—label-free line-pad-line; PDGF-BB—platelet-derived growth factor-BB; GDH—glucose dehydrogenase; Amp—amperometry; Biotin-Ab—biotinylated antibody; DTBSU—dithiobis(succinimidy undecanoate); AuWE—gold wire electrode; GO/MB—methylene blue loaded graphene oxide; PhA—physical adsorption; SH-PEG—thiol-modified polyethylene glycol; IL-6—interleukin-6; PSA—prostate specific antigen; IL-1β—interleukin-1β.

### 4.8. Prostate-Specific Antigen

Prostate cancer is the second most frequently occurring cancer in men worldwide and the fourth most commonly occurring cancer overall. There were 1.3 million new cases in 2018 [1,186]. Early prostate cancer usually shows no clinical symptoms and, therefore, requires efficient screening tests for its early diagnosis. Prostate-specific antigen (PSA), a serine protease of 30–34 kDa produced by epithelial prostatic cells, is the main biomarker for the diagnosis, screening and monitoring of patients with prostate cancer and eventually the first tumour biomarker approved by the Food and Drug Administration (FDA) [187]. The standard PSA cut-off is 0.125 nM (4 ng mL^−1^). However, with this cut-off, only 20.5% of the prostate cancer cases are tested positively and nearly 80% of this cancer cases are missed. Prostate cancer is often suspected when the PSA concentration is in the “diagnostic grey zone” from 4 to 10 ng mL^−1^ PSA [188,189].

There are clinically approved immunoassays for PSA detection [190,191] and numerous electrochemical aptasensors aiming at inexpensive and robust ways of PSA electroanalysis have been reported and intensively overviewed [192,193,194,195,196,197,198,199]. Here, we update only few very recent reports.

The aptasensor based on the hemin-functionalised graphene-conjugated palladium nanoparticles (H-Gr/PdNPs) deposited on GCE showed a linear response to PSA in the concentration range from 0.025 to 205 ng mL^−1^, with a LOD of 8 pg mL^−1^. The sensor was used for quantitation of PSA in spiked serum samples, giving recovery rates ranging from 95.0 to 100.3%. Hemin acted both as a protective agent and as an in-situ redox probe. PdNPs provided numerous binding sites for immobilisation of biotinylated DNA via coordinative binding between Pd and amino groups of DNA’s bases. The PSA aptamer was immobilised via biotin-streptavidin interactions [200].

In another study, the aptasensor was constructed by immobilisation of a thiolated DNA aptamer onto AuNPs/fullerene C60-chitosan-IL/MWCNT/SPCE. The aptasensor determined PSA by EIS and DPV in the range of 1 to 200 pg mL^−1^ and LOD of 0.5 pg mL^−1^ and 2.5 to 90 ng mL^−1^ and LOD of 1.5 ng mL^−1^, respectively. Analysis of the PSA in serum samples obtained from patients with prostate cancer was performed [201].

A simpler modification of SPCE with AuNPs and a thiolated aptamer combined with voltammetric analysis of signals from [Fe(CN)_6_]^3−/4−^ resulted in the linear response to PSA in the concentration range from 31.3 fM to 6.25 nM (1 pg mL^−1^ to 200 ng mL^−1^) PSA and LOD of 2.41 fM (0.077 pg mL^−1^) [202]. The aptasensor operated in undiluted human serum, with a negligible interference from BSA, IgG and Hb, and its performance was comparable to ELISA’s.

In a nanocomposite design, a thiolated aptamer was co-immobilised with MCH on the nanocomposite-modified GCE; a nanocomposite was: graphene quantum dots-chitosan-nafion-IL; MWCNTs-graphene-IL and polypyrrole-MoS_2_-IL-AuPt NPs [203]. Sensor’s SWV responses processed by a computerised monitoring system (SACMES) linearly changed with the PSA concentration in the 15.6 fM–10.9 pM (0.5–350 pg mL^−1^) range and LOD was 4.38 fM (0.14 pg mL^−1^). Neuron-Specific Enolase, HSA, Hb, thrombin, IgG and lysine did not interfere with the PSA detection, and the sensor operated well in serum [203]. A magnetic GCE modified with rGO/Fe_3_O_4_/Cu_2_O nanosheets was used as a transducer in another nanocomposite ratiometric aptasensor, electroactive Cu_2_O NPs deposited onto the surface of rGO/Fe_3_O_4_ NSs acting as reference tags (Figure 7). The aptamer-modified Ag@resorcinol-formaldehyde (RF) NPs were assembled on the magnetic electrode surface. After reaction with PSA, NPs were removed from the electrode surface and then worked as electroactive Ag nanodot detection tags. Aptasensor’s linear range was from 0.313 pM to 3.13 nM (0.01 to 100 ng mL^−1^) PSA, with LOD of 194 fM (6.2 pg mL^−1^). The aptasensor could operate in human serum with a low interference from α-fetoprotein, carcionoembryonic antigen, cysteine, GSH, and tryptophan [204].

### 4.9. Platelet-Derived Growth Factors

Platelet-derived growth factors (PDGF) are disulphide-bond stabilised heterodimeric peptides existing in a number of isoforms including PDGF-AA, PDGF-BB, and PDGF-AB. Among them, PDGF-BB is a cancer-related protein. It contributes to proliferation, survival, and motility of connective tissue and some other types of cells by initiating signalling via two receptor tyrosine kinases: α- and β-receptors. PDGF overexpression is linked to malignancies and diseases characterised by excessive proliferation of cells, such as atherosclerosis or fibrosis. PDGF play at least three roles that can lead to the tumour growth: autocrine stimulation of cancer cells, stimulation of angiogenesis, and control of tumour interstitial pressure, and are considered as prognostic biomarkers of numerous cancers [205]. Its cancer cut-off value is 6.87 × 10^−11^ M [206].

Electrochemical aptamer-based biosensors for PDGF-BB were numerously reported [207]. They exploit a variety of design approaches, exemplified by sandwich assays based on MOS_2_/carbon aerogel composites [208], aptamer-functionalised multidimensional hybrid conducting-polymer plates [209], carbon-based nanocomposites with aptamer-templated silver nanoclusters [210]. A simple in design label-free aptasensor constructed by immobilisation of a thiolated PDGF-binding aptamer and MCH on a gold electrode and using a [Fe(CN)_6_]^3−/4−^ redox indicator linearly responded to PDGF concentrations ranging from 1 to 40 nM (DPV analysis) [211]. Another simple approach suggested the electrochemical aptamer beacon design for the detection of PDGF biomarker directly in blood serum. Alternating current voltammetry was used to monitor interactions between MB-labelled aptamer immobilised on a gold electrode and PDGF [212]. The aptasensor detected the BB variant of PDGF at 1 nM directly in unmodified, undiluted blood serum and at 50 pM in serum diluted 2-fold with an aqueous buffer solution. The aptasensor was well suited for using in portable microdevices [212].

Selected examples of electrochemical aptasensors for PDGF, developed since 2015, are presented in Table 4. Here, we discuss only two most recent examples published since 2018.

In the first example, a Y-shaped DNA probe target-triggered amplification strategy was used for PDGF-BB detection in human serum [213]. A thiolated hairpin aptamer H1 co-immobilised with MCH on the AuNP-selenium-doped MWCNTs-graphene-modified GCE. After PDGF-BB addition, aptamer H1 bound with it and triggered the catalytic assembly of two other hairpin DNAs to form the G-quadruplex Y-junction DNA structures, which released PDGF-BB to bind again the intact aptamer and initiate another assembly cycle. Next, G-quadruplex/hemin complexes were formed when hemin was added; the former generated a substantially amplified current output. The DPV response of the aptasensor to PDGF-BB displayed a linear range from 0.1 pM to 10 nM with LOD of 27 fM. Thrombin, IgG, and PSA did not interfere [213].

In the second example, a flexible, three-dimensional carbon nanoweb (3D-CNW)-based aptamer platform was designed for PDGF-BB detection [214]. Poly(acrylonitrile) nanowebs (NWs) were used as a template for the overall 3D structure. A silver paste was screen-printed on both ends of the nanowebs and subjected to the chemical vapour deposition process using the Cu powder. Cu was then etched to generate carbon bulges on the surface of the 3D-CNW. The aminated DNA aptamer and 4-(4,6-dimethoxy-1,3,5-triazin-2-yl)-4-methylmorpholinium chloride) (a condensing agent) were coupled to the 3D-CNW surface of the transducer, by covalent binding between the amino group of the aptamer and the carboxyl groups of CNW. The analytical performance tested with a liquid-ion gated FET-type setup exhibited LOD of 1.78 fM, also in FBS, and the aptasensor response to PDGF was not compromised by calmodulin, adenosine triphosphate, and BSA [214].

**Table 4 sensors-21-00736-t004:** Analytical performance of selected electrochemical aptasensors for PDGF.

Strategy	Technique	LOD, M	Interference Studies	Ref.
Sandwich assay based on MOS_2_/carbon aerogel composites	DPV	0.3 × 10^−12^	BSA, CEA, Hb, IgG	[208]
Structure-switching hairpin probe	CV	2.67 × 10^−12^ (0.08 ng mL^−1^)	PDGF-AA, PDGF-AB, IgG, BSA	[215]
VS2-GR coupled with Exo ΙΙΙ-aided signal amplification leaf like VS2 nanosheets	DPV	0.03 × 10^−12^	BSA, IgE, Thr, CEA	[216]
	DPV	0.4 × 10^−12^	Hb, Thr, BSA	[217]
A background current-eliminated Apt sensing platform	ACV	0.334 × 10^−12^ (10 pg mL^−1^)	PDGF-AA, PDGF-AB	[218]
Fe_3_O_4_@3D-rGO@plasma-polymerised (4-vinyl pyridine) nanocomposite	EIS	0.98 × 10^−12^ (29.4 pg mL^−1^)	Thr, IgG, Lyz, BSA	[219]
Co_3_(PO_4_)_2_ BSA-based aptasensor	EIS	0.123 × 10^−12^ (3.7 pg mL^−1^)	BSA, IgG, Lyz, Thr, PDFG-AA, PDGF-AB	[220]
Exo ΙΙΙ-aided signal amplification strategy	DPV	20 × 10^−15^	Hb, BSA, IgG, CEA, BSA	[221]
Apt-functionalised MHCPP	FET	1.78 × 10^−15^	ATP, BSA, Cal, PDGF-AA	[209]
Apt based EGFET sensor RCA	EGFET	8.8 × 10^−12^	Not shown	[222]
Apt based dual signal amplification strategy using hydroxyapatite NPs	SWV	1.67 × 10^−15^ (50 fg mL^−1^)	AFP, CEA, IgG, HER2	[223]
Sandwich Ab-Apt labelled ALP	SWV	1.67 × 10^−15^ (50 fg mL^−1^)	AFP, CEA, IgG, p53, HER2	[224]
EXPAR	DPV	52 × 10^−15^	PDGF-AA, PDGF-AB	[225]
Carbon-based nanocomposites with Apt-Ag-NCs	EIS	26.5 × 10^−15^ (0.82 pg mL^−1^)	BSA, Thr, Lyz, IgG	[210]
Sandwich sensing system on 3D-IHC	Amp	1.2 × 10^−15^ (0.03 pg mL^−1^)	AA, UA, Gly, Glu	[226]
Aptasensor based on new structure of GNPs containing α-CD	SWV	0.52 × 10^−9^	BSA, PSA, HSA, p53	[227]
Se-doped MWCNTs-Gr, Hem/G-quadruplex and Y shaped DNA-aided target-triggered	DPV	27 × 10^−15^	Thr, IgG, PSA	[213]
Apt-Functionalised 3D CNWs	FET	1.78 × 10^−15^	Cal, ATP, BSA	[214]

Abbreviations: BSA—bovine serum albumin; CEA—carcinoembryonic antigen; Hb—haemoglobin; IgG—immunoglobulin G; Cal—calmodulin; AA—ascorbic acid; UA—uric acid; Gly—glycine; Glu—glucose; PDGF-AA—platelet-derived growth factor AA; PDGF-AB—platelet-derived growth factor AB; Exo III—Exonuclease III; IgE—immunoglobulin E; Thr—thrombin; Lyz—lysozyme; HER2—human epidermal growth factor receptor 2; AFP—alphafetoprotein; p53—cellular tumour antigen; SWV—Square Wave Voltammetry; EIS—electrochemical impedance spectroscopy; DPV—Differential Pulse Voltammetry; Amp—amperometry; FET—field-effect transistors; EGFET—extended-gate field-effect transistors; RCA—rolling circle amplification; AgNCs—silver nanoclusters; EXPAR—proximity hybridisation-induced isothermal; ACV—alternating current voltammetry; Apt-aptamer; Ab—antibody; HSA—human serum albumin; α-CD—Alpha-Cyclodextrin; ALP—alkaline phosphatase, PSA—prostate specific antigen; MHCPP—multidimensional hybrid conducting-polymer plate; MWCNTs—multiwalled carbon nanotubes; Gr—graphene; Hem—hemin; 3D CNWs—Three-Dimensional Carbon Nanowebs; ATP—Adenosine triphosphate; 3D-IHC—three dimensional inorganic hybrid composite.

### 4.10. α-Fetoprotein (AFP)

Alpha-fetoprotein (AFP) is a multifunctional glycoprotein of ca. 70 kDa with a dual regulatory role in cancer and fetal activity [228]. The serum content of AFP can be related to liver carcinomas, and APF is considered as an important diagnostic biomarker of hepatocellular carcinoma (HCC)–one of the most malignant tumours, representing 4.7% of all cancers. It is also the sixth most common cancer in the world and the third leading cause of cancer deaths in both sexes worldwide [1]. In serum of a healthy human, the AFP concentration does not exceed 25 ng mL^−1^, but it increases to 500 ng mL^−1^ in nearly 75% of HCC patients [229]. A concentration of 20 ng mL^−1^ is considered as a pathological threshold for AFP in clinical samples [230].

Numerous examples of immunosensors for AFP have been reported [231]. Though the electrochemical AFP aptasensor field is less populated, we selected some interesting results reported during last five years (Table 5). Considering the 20 ng mL^−1^ pathological threshold for AFP detection, all reported aptasensors’ LODs fully meet the clinical requirements, though in many cases samples apparently should be excessively diluted prior analysis.

Among those, graphene-based aptasensors have gained particular attention thanks to such properties of graphene as its large specific surface area, high carrier mobility and electrical conductivity, flexibility, and optical transparency. A simple label-free voltammetric aptasensor based on thionine/rGO/AuNPs-modified SPCE allowed LOD of 0.05 μg mL^−1^ AFP and the response linearity range from 0.1 to 100.0 μg mL^−1^ AFP [232]. A thiolated aptamer was immobilised on AuNPs, and the change of thionine redox activity upon AFP binding was DPV-registered. AFP analysis could be performed in human serum and such proteins as BSA, HSA, IgG, and IgE did not interfere [232]. In another simple design, NH_2_-aptamer was attached to COOH-GO deposited onto GCE through the EDC/NHS coupling reaction [233]. To prevent non-specific adsorption, the electrode surface was treated with BSA. The CV responses from ferro/ferricyanide were linearly proportional to the AFP concentration, within the 0.01 to 100 ng mL^−1^ range, with LOD of 3 pg mL^−1^. The aptasensor worked satisfactory in human serum, and PSA and carcinoembryonic antigen did not interfere [233].

A more complicated voltammetric aptasensor exploited the Prussian blue nanoparticle (PBNP)—labelled aptamer immobilised on GO nanosheets deposited onto a gold disk electrode. The AFP binding to the aptamer caused dissociation of PBNP, while in the presence of DNase I, the formed AFP-aptamer complex was cleaved to release AFP, which could again react with the PBNP–labelled aptamer, inducing the AFP recycling. A DPV response changed linearly within the 0.01 to 300 ng mL^−1^ AFP range, with LOD of 6.3 pg mL^−1^. AFP was detected in human serum, with a negligible interference from neuron-specific enolase, carcinoembryonic antigen, CA125, human chorionic gonadotropin, and PSA [234].

Nanomaterial-electrode modifications are also intensively explored to improve the sensitivity of the ferricyanide-based assays for AFP. Immobilisation of a thiolated aptamer on spindle-shaped gold nanostructures (ssAuNSs) electrodeposited on a gold electrode (with MCH as a backfilling agent) allowed the AFP detection within the 0.005 to 10 ng mL^−1^ range and LOD of 0.23 pg mL^−1^ [235]. IgG, carcinoembryonic antigen, BSA, and HSA did not interfere with analysis of AFP, and the AFP aptasensor was validated in real samples supported by ELISA tests as a reference [235]. Another nanomaterial strategy suggested immobilisation of a DNA aptamer on 3D-nitrogen-doped nanoporous carbon nanomaterials (N-nCN), obtained by pyrolysis (calcination) of green bristle grass and deposited on the gold electrode [236]. The AFP binding produced the DPV signal within the 0.1 pg mL^−1^ to 100 ng mL^−1^ range, with LOD of 60.8 fg mL^−1^. The aptasensor detected AFP in cancer patients’ serum samples with low interference from BSA, IgG, and other cancer-related proteins [236].

Alternative electrode modification strategies include emerging materials and less conventional matrices. A DNA aptamer was immobilised on a guanosine-pyridine-4-boronic acid-KCl hydrogel deposited on gold, and the AFP binding detected by EIS with [Fe(CN)_6_]^3−/4−^ allowed the 0.001 to 0.5 ng mL^−1^ AFP detection and LOD of 0.51 pg mL^−1^ [237]. The gold electrode functionalised with the metal organic framework—Cu ions guided MIL-96 octadecahedron crystals—used for the aptamer immobilisation exhibited LOD of 0.12 pg mL^−1^ AFP and a linear response within the 1 to 500 pg mL^−1^ range [238].

Redox labelling of the aptamer is another simple yet efficient signal acquisition strategy. An electronic DNA-hairpin beacon, one of the simplest nanoswitching architectures [239] was designed for 8.76 pg mL^−1^ detection of AFP; a thiolated and MB-labelled aptamer being immobilised on a gold electrode [240]. A more complicated design of the bi-directionally amplified ratiometric aptasensor exploited exonuclease-assisted target recycling and a ferrocene-labelled DNA capture probe (Fc-DNA) [241]. Fc-DNA was hybridised to the thiolated aptamer (DNA1) immobilised on gold, and AFP–aptamer binding resulted in dissociation of Fc-DNA from the hybrid. The process of RecJf exonuclease cleavage allowed further recycling of AFP and more Fc-taged DNA released, accompanied by more DNA1 exposed on the electrode surface for hybridisation with a MB-labelled DNA probe. This aptasensor’s LOD reached 0.27 fg mL^−1^, and AFP was detected in human serum, with no interference from thrombin, carcinoembryonic antigen, IgG, and PSA [241]. Another complex redox labelling strategy was proposed for a sandwich aptasensor design, with AFP captured on the thiolated-aptamer-modified gold electrode and covalently labelled by the Michael addition reaction with the maleimide-functionalised DNA probe [242]. The formed aptamer/AFP/DNA sandwich architecture was subjected to the hybridisation chain reaction induced with the DNA probe and kinetically trapped two MB-labelled hairpin DNA. MB voltammetric signals from those changed linearly for 0.1 pg mL^−1^ to 100 ng mL^−1^ AFP, and LOD was 0.041 pg mL^−1^ [242].

To prevent biofouling of the aptamer surface, a thiolated aptamer was also co-immobilised with a zwitterionic peptide as an antifouling agent, and the AFP binding was voltammetrically monitored with ferricyanide as a redox probe [243]. The aptasensor detected AFP at LOD of 3.1 fg mL^−1^, and HSA did not interfere with the sensor response.

**Table 5 sensors-21-00736-t005:** The analytical performance of selected electrochemical aptasensors for AFP.

Strategy	Technique	LOD, g mL^−1^	Interference Studies	Ref.
Apt/ZiP/Au	DPV	3.1 × 10^−15^	HSA, γ-globulin, Hb, ssDNA	[243]
Hairpin DNA-MB/Au		8.76 × 10^−12^	GP73, CEA	[240]
Apt/Thi/rGO/AuNPs/SPCE		5 × 10^−8^	BSA, HSA, IgG, IgE	[232]
Apt/GO/GCE	CV	3 × 10^−12^	PSA, CEA	[233]
Apt-PB NPs/GO/Au	DPV	6.3 × 10^−12^	NSE, CEA, MUC16, hCG, PSA	[234]
Apt/sAuNPs/Au		0.23 × 10^−12^	IgG, CEA, BSA, HSA	[235]
Apt-3D NMCNMs/Au		60.8 × 10^−15^	BSA, PSA, CEA, IgG, EGFR, MUC1, VEGF	[236]
Apt/Gs-Pyr BA-KCl HG/Au	EIS	0.51 × 10^−12^	Lyz, IgG, HSA, BSA, CEA	[237]
Apt/Cu MIL-96 OH/Au		0.12 × 10^−12^	CEA, IgG, Lyz, BSA, HSA	[238]
HCR—two DNA hairpins-MB/DNA probe/AFP/Apt/Au	DPV	0.041 × 10^−12^	BSA, IgG, IgE	[242]
MB-DNA-AuNPs/AFP/Fc-capture probe/MCH/DNA1/Au	ACV	0.27 × 10^−15^	Thr, CEA, IgG, PSA	[241]

Abbreviations: Au—gold electrode; Apt—aptamer; ZiP—zwitterionic peptide; DPV—differential pulse voltammetry; Hb–haemoglobin, ssDNA—single stranded DNA; CV—cyclic voltammetry; MB—methylene blue; AFP—alphafetoprotein; Thi—thionine; NSE—neuron-specific enolase; 3D NMCNMs—three-dimensional nitrogen-doped mesoporous carbon nanomaterials; GP73—golgi protein 73; CEA—carcinoembryonic antigen; BSA—bovin serum albumin; HSA—human serum albumin; IgE—immunoglobulin E; IgG—immunoglobulin G; GO—graphene oxide; rGO—reduced graphene oxide; AuNPs—gold nanoparticles; hCG—human chorionic gonadotropin; PSA—prostate specific antigen; EGFR—epidermal growth factor receptor; MUC1, MUC16—transmembrane mucins; VEGF—vascular endothelial growth factor; NPs—prussian blue nanoparticles; sAuNS—spindle-shaped gold nanostructure; Lyz—lysozyme; SPCE—screen printed carbon electrode; HCR—hybridisation chain reaction; GCE—glassy carbon electrode; EIS—electrochemical impedance spectroscopy; Fc—ferrocene; MCH—6-mercaptohexan-1-ol; ACV—alternating current voltammetry; Thr—thrombin; Gs-Pyr BA-KCl HG—guanosine-pyridine-4-boronic acid-KCl hydrogel; Cu MIL-96 OH—Cu ion guided MIL-96 octadecahedron.

### 4.11. Carcinoembryonic Antigen (CEA)

Carcinoembryonic antigen (CEA) is a heavily glycosylated 180 kDa cell-surface glycoprotein that is involved in cell adhesion and expressed during human foetal development. This oncofoetal protein is abnormally expressed in many human cancers, such as breast, lung, gastric, ovarian, thyroid, and pancreatic cancer, but is most commonly associated with colorectal cancer. It is abundantly expressed in approximately 95% cases of colorectal cancer [244,245], which is the third most prevalent cancer and the second cause of cancer fatality worldwide [1]. The detection of CEA cancer biomarker thus became essential for differential diagnosis, condition monitoring, and therapeutic assessment of colon cancers with a cut-off of 5 ng mL^−1^ [246].

Over the last five years, an impressive number of electrochemical aptasensors for the CEA detection have been developed and we summarise the most recent examples in this section and Table 6.

A variety of nanomaterials and nanocomposites was used both for the aptamer immobilisation and labelling: graphene-based nanomaterials [247,248,249,250,251,252,253,254,255,256], nanocomposite Zr metal–organic frameworks with Ag nanoclusters [257], and avidin-gated mesoporous silica nanoparticles [258].

A multi-signal-amplified voltammetric aptasensor exploiting the lead-based metal organic framework (Pb-MOF) signal marker enabled a 0.333 pg mL^−1^ CEA detection in serum, within the 0.001–100 ng mL^−1^ range [254]. A DNA aptamer-modified gold electrode was combined with the Pb-MOF-labelled dendritic DNA nanostructures formed in the hybridisation chain reaction. The target substitution reaction released internal Pb^2+^ ions to be detected by SWV. The aptasensor results were comparable with ELISA [254].

Zheng and colleagues proposed a strategy of designing a biosensor for CEA detection based on copper silicate integrated with nitrogen-doped magnetic carbon microtubes (NCMTs@Fe_3_O_4_@Cu silicate). For this purpose, concanavalin A (ConA) was bound to the previously synthesised NCMTs@Fe_3_O_4_@Cu silicate composite by chelating with copper ions. ConA acted as an affinity probe for the selective capturing of CEA. Next, gold nanoclusters (AuNCs) labelled with the DNA aptamer interacted with CEA, creating a sandwich. The electrochemical response of this aptasensor was generated by the electrochemical oxidation of AuNCs to AuCl_4_^−^. It detected CEA within a linear range from 0.03 to 6.00 ng mL^−1^ and LOD of 5.38 pg mL^−1^. The analytical performance of the assay in human serum gave satisfactory results [259].

Co-polymerisation of polydopamine and poly(sulfobetaine methacrylate) on GCE provided a matrix for a thiolated DNA aptamer immobilisation via the Michael addition reaction. An electrochemical sensing performance of the aptasensor was investigated [Fe(CN)_6_]^3+/4+^. It displayed a linear range of 0.01 to 10 pg mL^−1^ and LOD of 3.3 fg mL^−1^ and operated in human serum samples [260].

The Cu_3_(PO_4_)_2_ nanoflowers/GO nanocomposites (Cu_3_(PO_4_)_2_NF@GO) represented another platform for the biotinylated DNA aptamer immobilisation, via the biotin—streptavidin linkage [255]. The reaction of molybdophosphate precipitation from Cu_3_(PO_4_)_2_NF@GO and molybdate (MoO_4_^2−^) electrochemistry were used as an indicator. CEA binding to the aptamer hindered the molybdophosphate formation and decreased the electrochemical signal from molybdite, changing linearly with the CEA logarithmic concentration in the range from 10 fg mL^−1^ to 500 ng mL^−1^ (LOD of 2.4 fg mL^−1^) [255].

A conductive strip paper-based aptasensor detected CEA in the 0.77 to 14 ng mL^−1^ range, with LOD of 0.45 ng mL^−1^ and 1.06 ng mL^−1^ in buffer and human serum samples [256]. The sensing layer represented graphene and poly (3,4-ethylenedioxythiophene): poly(styrenesulfonate) (PEDOT:PSS) nanocomposite, (3-aminopropyl) triethoxysilane (APTES), succinic anhydride (SA), DNA aptamers, and BSA. APTES provided amine groups on the electrode surface and SA-carboxyl groups for attachment of the aminated aptamer. Ferricyanide was a redox probe for the impedimetric detection of CEA [256].

A toehold-aided DNA recycling voltammetric aptasensor exploited a SH-ssDNA probe hybridised with DNA1 and DNA2 to form a 3’-toehold domain [261]. Partially hybridised SH-ssDNA immobilised on gold (MCH used for surface backfilling) reacted with a complex of CEA-aptamer and a free DNA probe. The latter was hybridised with SH-ssDNA and triggered toehold-aided DNA recycling. Finally, MB-labelled DNA3 hybridised to SH-ssDNA, and the MB signal was proportional to the logarithmic concentration of the assayed CEA (the 0.1 to 50 ng mL^−1^ linear range, LOD of 20 pg mL^−1^). This aptasensor operated in spiked serum, urine, and saliva [261].

**Table 6 sensors-21-00736-t006:** The analytical performance of selected electrochemical for CEA.

Strategy	Technique	LOD, g mL^−1^	Interference Studies	Ref.
Glu ox–Fc nanoporous AuNSs/CEA/MCH/Apt/Au	DPV	0.45 × 10^−12^	HSA, human IgG, mouse IgG	[262]
MCH/Apt/AuNPs-Hem/GCE		40 × 10^−15^	HSA, Thr, Lyz, Ins	[247]
Apt-C-PPy MNTs/IDMA	FET	1 × 10^−15^	Thr, BSA, DP, AA, UA	[263]
AuNPs-Apt 2-CEA-Apt1/Au	Amp	0.899 × 10^−12^ (5 fM)	PSA, BSA	[264]
H Apt +AuNRs Gr-strp NM/GCE	DPV	1.5 × 10^−12^	Myo, Fer, IgG	[248]
H1 + H2 + MnTMPyP/Pt-Pd–Apt2/CEA/BSA/Apt1/AuNPs/GCE	EIS	0.03 × 10^−12^	Thr, rabbit IgG, AFP, PSA	[265]
UiO-66-AgNCs-Apt/Au	EIS DPV	8.88 × 10^−12^ 4.93 × 10^−12^	AA, MUC1, Thr, IgG,	[257]
BSA/Apt/AuNPs/Gr-MoSe_2_ hybrid/GCE	DPV	0.03 × 10^−12^	HSA, IgE, AFP, Thr, LDL	[249]
MoO_4_^2−^/AuNRs-Apt/Ab-CEA/GO/GCE	SWV	0.05 × 10^−12^	AFP, MUC16, IgG	[250]
CuMOFs-PtNPs-Apt2-hemin-GOx/CEA/BSA/Apt1/AuNPs/GCE	EIS	0.023 × 10^−12^	Thr, IgG, AFP, PSA	[251]
Auxiliary DNA/Pb^2+^/Substrate chain-MB /DNAzyme: Hairpin-CEA/GrQD-IL-Naf/GCE	DPV	0.34 × 10^−15^	BSA, PSA, MUC1	[252]
Compl. strands: DNA1 and DNA2/Hairpins: Apt1 + Apt2/SPAuE		0.9 × 10^−12^	IgE, Thr, HSA, Gly, Myo, PSA	[266]
HRP/ConA/CEA/DNA Apt + MCH/Au		3.4 × 10^−9^	BSA, HSA, ɤ-globulin, AFP, CRP	[267]
Apt/NiCoPBA/Au	EIS	0.74 × 10^−15^	BSA, HER2, IgG, PSA, VEGF, MUC1	[268]
MSiNPs–MB–Av/IBNc /CEA/BSA/Apt/AuNPs/SPCE	DPV	280 × 10^−15^ (buffer) 510 × 10^−15^ (HS)	Not shown	[258]
DMP DTPs HCR/Au		18.2 × 10^−15^	Hb, PSA, Thr	[269]
EA/Apt/ Hem-GO-MWCNs/GCE		0.82 × 10^−15^	Ins, Ua, Glu, Arg, Gly, HSA	[253]
Au-SiO_2_ Janus nanoparticles–Apt/Av-Fe_3_O_4_@SiO_2_ NanoCaptors/SPCE	Amp	210 × 10^−12^	Thr, IgG, HSA	[270]
HRP-Cu_3_(PO_4_)_2_ hybrid nanoflowers-AuNPs-Apt2-BSA/EA/BSA/Apt1/Hemin-rGO-AuNPs/GCE	DPV	29 × 10^−15^	Thr, Cys, Hb, PSA,	[271]
sDNA-Pb-MOF/H1 and H2-HCR/DNA/Apt/Au	SWV	0.33 × 10^−12^	Myo, BSA, MUC1, PSA	[254]
Apt/PDA + PSBMA/GCE	DPV	3.3 × 10^−15^	Lyz, ɤ-globulin, MB, IgG, BSA, HSA	[260]
MoO_4_^2−^/CEA/Apt-Cu_3_(PO_4_)_2_ hybrid nanoflowers + GO composites/GCE	SWV	2.4 × 10^−15^	PSA, Thr, Hb	[255]
BSA/Apt/SA/APTES/Gr-PEDOT:PSS/PSE	EIS	0.45 × 10^−9^ (buffer) 1.06 × 10^−9^ (HS)	BSA, PSA, Ins	[256]
DNA3-MB/CEA/Apt/MCH/DNA2/DNA1/SH-ssDNA/Au	SWV	20 × 10^−12^	BSA, PSA, CD86, EpCAM	[261]
AuNCs-Apt/CEA/ConA/NCMTs-Fe_3_O_4_-Cu silicate/Au	DPV	5.38 × 10^−12^	Hb, PDGF, Lyz, Thr, BSA, Cys, IgG, PSA	[259]

Abbreviations: CEA—carcinoembryonic antigen; Glu ox—glucose oxidase; AuNSs—gold nanospheres; Fc—ferrocene; MCH—mercaptohexan-1-ol; Au—gold electrode; Apt—aptamer; DPV—differential pulse voltammetry; AuNPs-Hem-Gr—ternary nanocomposite of gold nanoparticles, hemin and graphene nanosheets, GCE—glassy carbon electrode; IDMA—interdigitated microelectrode array; C-PPy MNTs multidimensional conducting-polymer (3-carboxylate polypyrrole) nanotubes; DPV—differential pulse voltammetry; FET—field-effect transistors; AuNRs—gold nanorods; Gr-strp NM—graphene-streptavidin nanomatrix; H1, H2—two DNA hairpins; MnTMPyP—manganese (III) meso-tetrakis(4-N-methylpyridiniumyl)-porphyrin; Ab—antibody; Pt-Pd nanowires—platinum-palladium nanowires; Amp—Amperometry; HSA—human serum albumin; human IgG, mouse IgG, rabbit IgG—immunoglobulins G; Amp—amperometry; Thr—thrombin; Lyz—lysozyme; Ins—insulin; BSA—bovine serum albumin; DP—dopamine; HS—human serum; AA—ascorbic acid; UA—uric acid; PSA—prostate specific antigen; Myo—myoglobin; Fer—ferritin; AFP—alphafetoprotein; MUC1, MUC16—transmembrane mucins; MOF—Metal–organic framework; IgE—immunoglobulin; LDL—low density lipoprotein; Gly—glycine, CRP—C reactive protein; HER2—human epidermal growth factor receptor 2; VEGF—vascular endothelial growth factor; Hb—haemoglobin; Ua—urea; Glu—glucose; Ar—arginine; Cys—L-cysteine; MB—methylene blue; CD86—Cluster of Differentiation 86; EpCAM—Epithelial cell adhesion molecule; PDGF—platelet-derived growth factor; ConA—concanavalin; NCMTs-Fe_3_O_4_-Cu silicate—hierarchical copper silicate integrated with nitrogen doped magnetic carbon microtubes composite; PSE—paper-strip electrode; PEDOT:PSS—poly (3,4-ethylenedioxythiophene): poly(styrenesulfonate); SA—succinic anhydride; APTES—(3-Aminopropyl)triethoxysilane; PDA-PSBMA—Copolymer: Polydopamine and poly(sulfobetaine methacrylate); HCR—hybridisation chain reaction; SPCE—screen-printed carbon electrode; EA—ethanoloamine; Hem-GO-MWCNs—ternary nanocomposite of hemin, graphene oxide and multiwalled carbon nanotubes; HRP—horseradish peroxidase; MSiNPs—mesoporous silica nanoparticles; Av—avidin; IBNc—iminobiotin nanocarrier; SPAuE—screen printed gold electrode; DMP—dual-function messenger probe; DTPs—DNA tetrahedron probes; GrQD-IL-Naf—graphene quantum dot-ionic liquid-nafion composite; NiCoPBA—bimetallic Ni-Co Prussian blue analogue nanocubes; CuMOFs—Cu-based metal-organic frameworks functionalised; PtNPs—Pt nanoparticles; UiO-66—AgNCs—nanocomposite of zirconium metal–organic framework embedded with silver nanoclusters.

## 5. Multiplex Electrochemical Aptasensor Platforms for Several Cancer Biomarker Detection

Simultaneous detection of multiple biomarkers allows efficient screening, diagnosis, and monitoring of neoplastic diseases. Despite significant recent advances in the development of aptasensors, most research is still focused on single-analyte detection. However, clinical trials request the accurate and sensitive multianalyte detection, providing a much more detailed information on a range of biomarkers and cancer state.

To address the challenge of saving the time and cost of multiplex analysis, a voltammetric dual sensor for the parallel and continuous detection of MUC1 and CEA was constructed by using a thiolated SH-DNA probe attached to the MCH-blocked gold electrode surface [272]. The MB-labelled MUC1-specific and CEA-specific aptamer sequences were immobilised on the DNA-modified electrodes via their hybridisation with the SH-DNA probe in a way that MB was a signal-on probe for MUC1 binding and a signal-off probe for the CEA detection. The simultaneous analysis of two proteins showed the linear response to MUC1 within the 10 nM to 100 nM range (LOD of 0.6 nM) and from 30 ng mL^−1^ to 300 ng mL^−1^ CEA (LOD of 2.75 ng mL^−1^). Inverse trends in the current signal changes were observed with the increasing concentrations of MUC1 and CEA, generating an increment and a decrement in the current response, respectively. Tumour necrosis factor-α (TNF-α) and thrombin produced significantly lower responses. The aptasensor continuously monitored CEA (after the MUC1 binding) at LOD lowered then to 0.5 ng mL^−1^ [272]. Moreover, the aptasensor can be regenerated by rehybridisation of DNA2 or DNA3-MB after the parallel detection targets, or by introducing MUC1 and CEA and then rehybridising the MUC1 and CEA aptamers with linker DNA1 on the gold electrode. Therefore, it can be elucidated that this present strategy provides different protocols (parallel and continuous detection) to meet the requirements of analytical applications [272].

Another strategy is based on using multiple redox labels for simultaneous analysis of several proteins, such as an aptasensor exploiting two different integrated signalling probes (ISP)—two aptamers labelled with MB and Fc redox tags (sP1 and sP2) [273]. ISPs were immobilised onto the AuNP-modified GCE (blocked with MCH), and CEA and MUC1 binding resulted in a gradual decrease of both currents from MB and Fc detected by SWV. The linear range for the CEA detection was from 1 ng mL^−1^ to 1 µg mL^−1^ and for MUC1—from 5 nM to 1 µM. LODs for individual analysis of CEA and MUC1 were 0.517 ng mL^−1^ and 1.06 nM, while for parallel analysis they were 0.5274 ng mL^−1^ and 1.82 nM, respectively. HSA and HIgG did not interfere with CEA and MUC1 analysis. Comparing with the conventional independent signal probes for the simultaneous multi-analyte detection, the proposed ISP was more reproducible and accurate. This can be due to the fact that ISP in one DNA structure can ensure the completely identical modification condition and an equal stoichiometric ratio between sP1 and sP2, and, furthermore, the cross interference between sP1 and sP2 can be successfully prevented by regulating the complementary position of sP1 and sP2 [273].

Shell-encoded AuNPs: Au@Cu_2_O core-shell NPs and Au@Ag core-shell NPs were suggested as dual-analysis labels with redox potentials at −0.08 V and 0.26 V, correspondingly [274]. Two aminated DNA probes were immobilised via sulfamine bonds on the surface of GCE-containing sulfonic groups introduced by reductive electrolysis of 4-aminobenzenesulfonic acid, and AuNPs-Ag- and AuNPs-Cu_2_O-conjugated AFP and CEA aptamers were hybridised to both. AFP- and CEA-aptamer complex formations removed the AuNPs-labelled aptamers from the electrode surface, which resulted in the decreasing DPV signals from the shell-encoded AuNPs. The non-interfering and amplified DPV responses enable shell-encoded Au NPs to be an alternative electrochemical signal amplifier for dual screening of CEA and AFP. The duo-sensor showed linearity of responses for both biomarkers in the ranges from 1 pg mL^−1^ to 10 ng mL^−1^ AFP and from 5 pg mL^−1^ to 10 ng mL^−1^ CEA, with LODs of 1.8 pg mL^−1^ CEA and 0.3 pg mL^−1^ AFP. This aptasensor operated in blood samples and cellular extracts, with no interference from PSA, thrombin, IgG, L-cysteine, glutathione, arginine, histidine, valine, tryptophan and lysine. In comparison to the parallel single-analyte assays, shell-encoded Au NPs engineered electrochemical aptasensors offer multiplexing capability and show significant prospects in biomedical research and early diagnosis of diseases [274].

Simultaneous multiplexed detection of CEA and neuron specific enolase (NSE) was performed in a paper-based device produced by wax- and screen-printing, which enabled both the filtration and automatic injection of samples [275]. The device consisted of two integrated working carbon electrodes, which could simultaneously detect CEA and NSE in one sample by using two different redox modifications of the electrode. Each of the electrodes was modified either with amino-functionalised graphene-thionine-AuNPs or Prussian blue-PEDOT-AuNPs nanocomposites, to which either a thiolated CEA aptamer or a thiolated NSE aptamers were attached through the Au-S chemistry. CEA and NSE binding by the aptamers resulted in the linear drop of the DPV signals, at −0.25 V for thionine and at −0.02 V for Prussian blue, within the 0.01 to 500 ng mL^−1^ CEA and 0.05–500 ng mL^−1^ NSE concentration ranges. LODs of 2 pg mL^−1^ CEA and 10 pg mL^−1^ NSE were orders of magnitude lower than the cut-off serum values for lung cancer: 5 ng mL^−1^ CEA and 15 ng mL^−1^ NSE. A label-free electrochemical method was adopted, enabling a rapid simple point-of-care testing. The aptasensor device was used for clinical serum samples analysis, and BSA, uric and ascorbic acids, and Cyfra21-1 did not interfere [275].

Hyperbranched DNA-LaMnO_3_ perovskite (DNA-LMO) nanobiocomposites loaded with three different metal ions were used as redox probes for simultaneous discrimination of three different biomarkers of cancer, AFP, CEA, and PSA (Figure 8, left panel) [276]. Three aptamers were attached to a gold stirring rod and hybridised to DNA-(LMO-M)_n_ encoded probes. The encoded probes were prepared stepwise: (1) Three DNA strands complementary to the CEA, AFP, and PSA aptamers were extended by the TdT enzyme to form a long adenine (A)-rich ssDNA; (2) Pb^2+^, Cd^2+^, and Cu^2+^ ions were loaded onto the LMO nanoparticles then labelled with thymine (T)-rich ssDNA; (3) A-rich ssDNA hybridised with multiple DNA labelled LMO-M to get a set of encoded signal labels corresponding to each biomarker assay DNA-(LMO-M)_n_. After protein binding, the corresponding encoded labels were substituted by AFP, CEA, and PSA biomarkers, and SWV analysis of the supernatant detected the specific metal ion signals from each label encoding the particular protein in one run. The current response linearly increased with the protein concentrations in the range of 0.0001 to 100 ng mL^−1^, and LODs were 0.34 pg mL^−1^ AFP, 0.36 pg mL^−1^ CEA and 0.28 ng mL^−1^ PSA. Since the stirring rod can enrich many encoded probes containing many metal ions, multiplex signal amplification can be realised. Due to the enrichment and easy separation of the stirring rod, the signal-to-noise ratio was also obviously improved and thus to results in good sensitivity and accuracy. The results of electroanalysis of patients’ sera samples were comparable with those of ELISA [276].

Further, the possibility of a three-analyte detection–of p53 DNA, thrombin, and VEGF165—was shown with a three-channel carbon SPCE selectively modified with DNA probes specific for those analytes (Figure 8, right panel) [277]. SPCE were gold-plated and modified with a mixed SAM of azidoundecanthiol and mercaptoundecanol. Surface functionalisation with three different MB-labelled probes (p53 DNA- and protein-specific sequences) was performed by “click” chemistry assisted by electrocatalytically generated Cu(I). The reaction was carried out by sequential exposure of each azide-modified electrode to “click” solutions with a potential bias (−0.4 V) applied only to the selected electrode. The ACV responses of the electronic beacons formed on each of three working electrodes to their analytes linearly changed with the analyte concentrations, for 1.0 nM to 128 nM p53 DNA, 0.8 nM to 40 nM thrombin, and 0.1 nM to 5.0 nM VEGF165, respectively. LODs were 0.35 nM p53 DNA, 0.22 nM thrombin, and 0.014 nM VEGF165, and the aptasensor operated in fetal bovine serum [277].

## 6. Conclusions and Future Directions

It is clear that recent advances in the construction of electrochemical aptasensors for protein cancer biomarkers allow considering them as robust tools for efficient and reliable analysis of these proteins in physiological fluids, operating often at concentration levels well below the clinically required limits. Compared both to the optical and electrochemical immunoassay approaches, using aptamers as biorecognition elements makes the aptasensor development much more versatile and more easily compatible with nanomaterials and DNA nanotechnologies, both at the level of chemical modifications of the aptamer sequences and their enzymatic treatments [16]. Current achievements in nanoscience and colloidal particles production, such as production of NPs of the controllable size and properties, allow further optimisation of analytical systems and obtaining detectable signals for trace amounts of proteins [278,279], with this improving the aptasensor performance to the required levels of sensitivity. Further, the possibility of easy miniaturisation of the electrochemical aptasensor devices, short analysis time, and relatively low cost (compared to the antibody-based assays) make the electrochemical aptamer-based assays most promising for clinical analytical applications.

However, despite numerous scientific reports on excellently in-lab performing electrochemical aptasensors overviewed here and their apparent analytical and technological advantages, the number of practical applications of aptamers in clinical trials, such as for validation of liquid biopsies for diagnosis of cancers or for routine clinical quantification of already established serum biomarkers of cancer, is scarce. That occurs both due to the current prevalence of the optical ELISA-based approaches impeding the wide acceptance of electrochemical biosensing schemes in clinical practice and due to often complicated or commercially un-attractive solutions for the aptamer-electrode modifications and assay protocols. Insufficient long-term stability of bioelectronics chips, such as those based on thiolated aptamer immobilisations on gold electrodes, which are complicated and have low compatibility with automatic production electrode modifications with multiple nanomaterials and cumbersome assay protocols make upscaling of most of the suggested technological solutions unfeasible, and they do not yet approach the level of commercial kits for research and development, not mentioning the clinically validated device level. Currently, the most successful electrochemical platform for analysis of blood-circulating proteins is Abbott’s iStat Systems for acute disease monitoring and Roche cobas^®^ 6000 ECL analyser for tumour biomarkers, both exploiting the electrochemical ELISA, i.e., the immunoassay approach [52,280]. The next step may be the development of a similarly commercially attractive electrochemical aptamer-based platform for unambiguous analysis of protein signatures of the cancer type and cancer state.

Construction of practical aptasensors for multiplex POCT may be the right step to make electrochemical aptasensors commercially attractive. There were a number of strategies reported for the multiplex analysis (Section 5) based on either simultaneous use of several different aptamers in an array format or of several redox-active probes encoding each aptamer/protein binding, and their further adaptation to multichannel, multianalyte [281], electrode array [282], and label-free detection schemes [283] may further assist the market activation of electrochemical aptasensors for cancer diagnosis. Therewith, the limitations to overcome also include the on-chip reagent storage and decreasing the cost of the miniaturised aptasensors. For the latter, the cost per test is one of the significant value parameters in the POCT sensors development [284].

To sum up, the future development of integrated, portable, and automated electrochemical aptasensor devices for clinical diagnostics of cancer seems to be a realistic goal, and there are a critical number of excellent approaches reported in the scientific literature that may enable that. However, that would not happen without ultimate moving of the electrochemical aptasensors research from the purely electroanalytical to applied clinical applications, entering the translational medicine field.

## Figures and Tables

**Figure 1 sensors-21-00736-f001:**
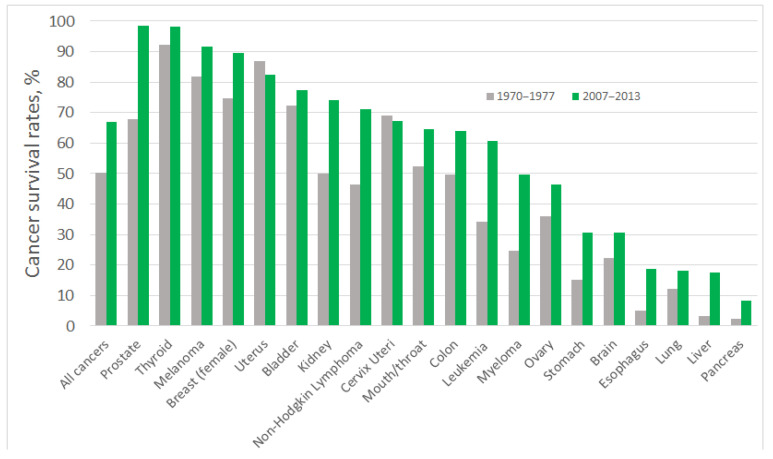
Five–year cancer survival rates in the USA, shown as the rate over the period 1970–1977 (grey bars) and 2007–2013 (green bars). The five–year interval indicates the percentage of people who live longer than five years following the diagnosis, data were taken from [7].

**Figure 2 sensors-21-00736-f002:**
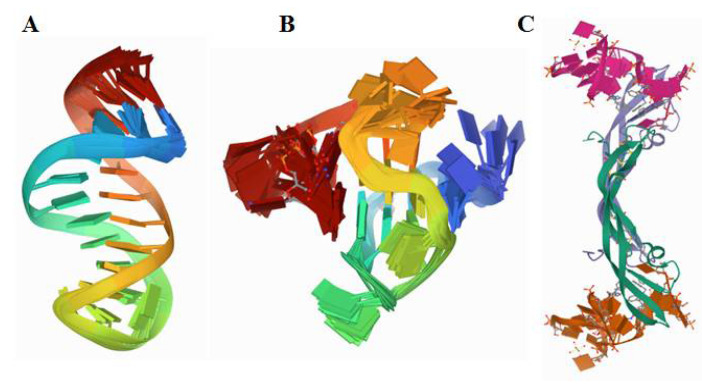
Structure of (**A**) a truncated 23-mer DNA MUC1 aptamer, image from the RCSB PDB (rcsb.org) of PDB ID 2L5K [18], (**B**) a VEGF aptamer with locked nucleic acid modifications, image from the RCSB PDB (rcsb.org) of PDB ID 2M53 [19], (**C**) human PDGF-BB protein in a complex with a modified nucleotide aptamer, image from the RCSB PDB (rcsb.org) of PDB ID 4HQU [20].

**Figure 3 sensors-21-00736-f003:**
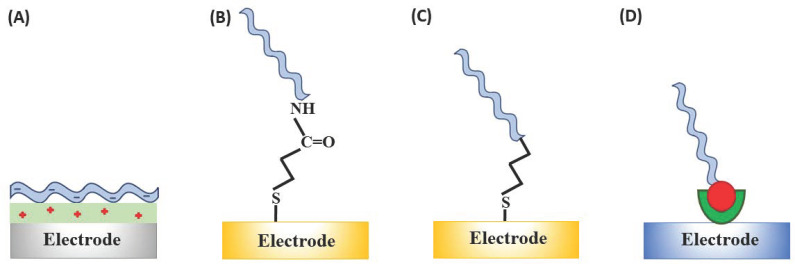
Immobilisation strategies for the aptasensor construction: (**a**) Physical adsorption of, e.g., negatively charged aptamer on the positively charged surface (e.g., polymer modified); (**b**) covalent attachment through the EDC/NHS ((1-ethyl-3-(3-dimethylaminopropyl)carbodiimide/N-hydroxysuccinimide) coupling between COOH-functionalised electrode surface and the amine-terminated aptamer sequence; (**c**) chemisorption on gold through the alkanethiol linker; and (**d**) through affinity (streptavidin-biotin) interactions.

**Figure 4 sensors-21-00736-f004:**
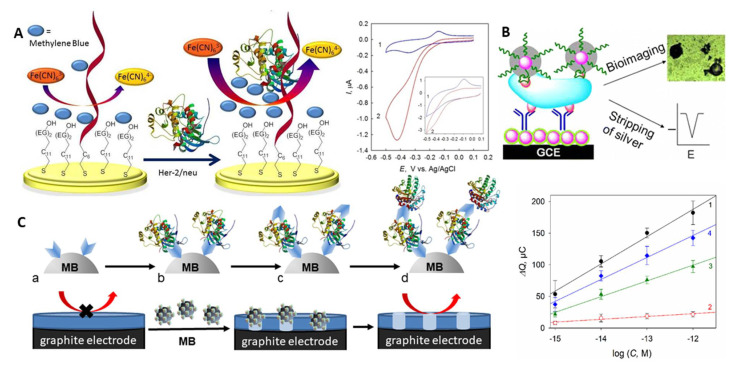
(**A**) Schematic representation of the electrocatalytic aptamer/PEG-based assay for HER-2/*neu* and voltammograms recorded with the aptamer/PEG-modified electrode in (1) MB and (2) MB and K_3_[Fe(CN)_6_] solutions, scan rate 0.1 V s^−1^ (inset: 5 V s^−1^) [103]. Specific binding sites and change of the aptamer conformation upon binding are not shown. (**B**) Silver-enhanced hybrid sandwich immunoassay with hydrazine-modified AuNP tags [105]. (**C**) Cellulase-linked sandwich immunoassay on MBs modified (**a**) with a capture antibody or an aptamer, (**b**) HER-2/*neu*, (**c**) a reporter antibody or aptamer and (**d**) cellulase, and dependence of the sensor response on the concentration of (1,3,4) HER-2/*neu* and (2) serum albumin for (1,2) Ab-(protein)-Ab-MBs, (3) aptamer-(protein)-Ab-MBs, and (4) aptamer-(protein)-aptamer-MBs sandwiches [107]. Copyright (2017) Wiley, copyright (2013) American Chemical Society and copyright (2019) Elsevier, reprinted with permissions.

**Figure 5 sensors-21-00736-f005:**
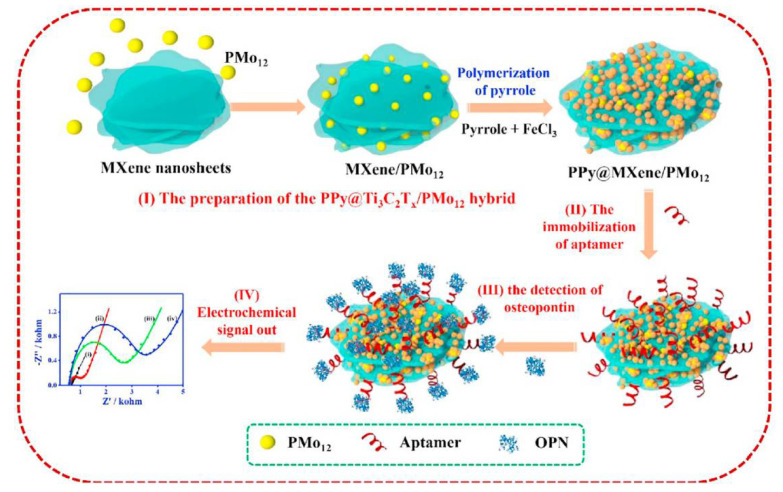
Schematic representation of the fabrication of the aptasensor for OPN based on the PPy@Ti_3_C_2_T_x_/PMO_12_ hybrid: (I) the preparation of PPy@Ti_3_C_2_T_x_/ PMo_12_, (II) the aptamer immobilisation, (III) the OPN detection, and (IV) the electrochemical signal read-out [135]. Copyright (2019) Elsevier, reprinted with permission.

**Figure 6 sensors-21-00736-f006:**
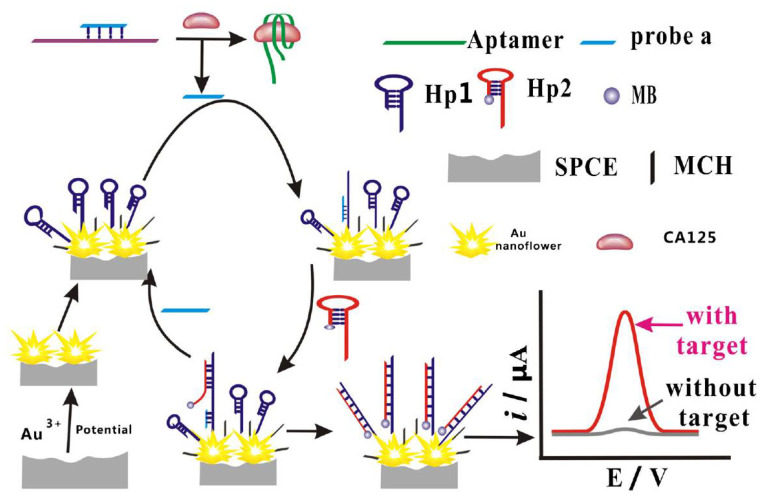
Schematic representation of the stepwise preparation of the biosensor for CA125 detection based on the use of flower-like gold nanostructures and target-triggered strand displacement amplification [160]. Copyright (2019) Springer, reprinted with permission.

**Figure 7 sensors-21-00736-f007:**
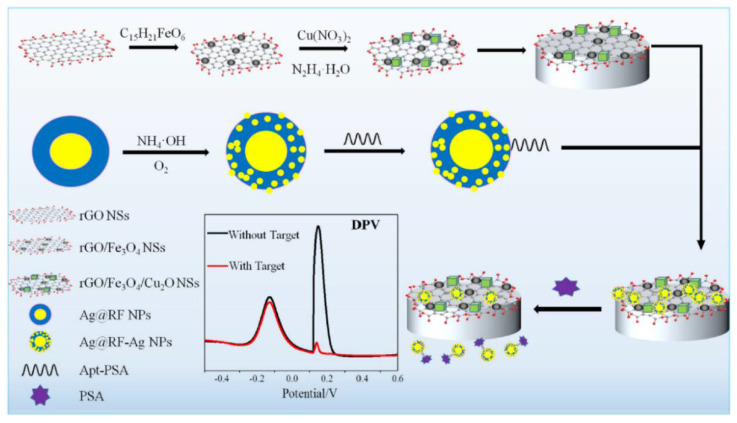
Schematic illustration of the electrochemical ratiometric aptamer-based assay for PSA [204]. Copyright (2020) Elsevier, reprinted with permission.

**Figure 8 sensors-21-00736-f008:**
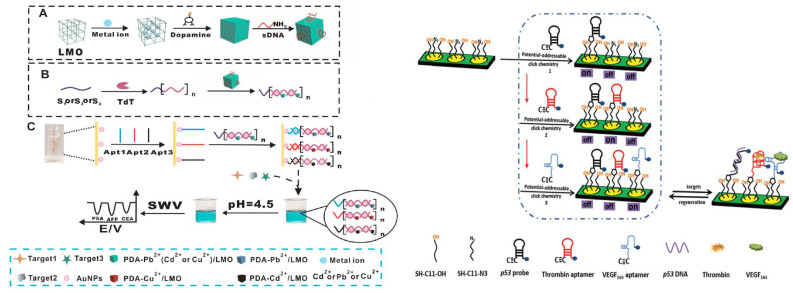
Aptasensors for multiplex detection of biomarkers (**left panel**) based on hyperbranched DNA-LaMnO_3_ perovskite (DNA-LMO) nanobiocomposites probes loaded with metal ions [276], Copyright (2021) Elsevier, reprinted with permission, and (**right panel**) based on a three-channel carbon SPCE selectively modified with DNA probes specific for three different analytes, reproduced from ref. [277] with permission from The Royal Society of Chemistry.

**Table 1 sensors-21-00736-t001:** Comparison of the methods of aptamers deposition on solid surfaces [76,77].

Immobilisation Strategy	Type of Interaction	Advantages	Disadvantages
Electrostatic adsorption	negatively charged aptamer on the positively charged surface	fast, simple, low-cost method	low packing density and control of orientation, random orientation of the aptamer causing layer instability under various conditions, such as changing the ionic strength of the buffer, pH, or other reagents, non-specific adsorption, low binding capacity and operational stability
Covalent attachment	EDC/NHS coupling between COOH-functionalised electrode surface and the amine-terminated aptamer sequence	stability (bond can be broken under extreme conditions), well-ordered layer, high degree of orientation control and thickness of electrode surface, single-point attachment of the probe at the thiol group end-point	difficult preparation method, high cost
Chemisorption	involves chemical bond between the probe and the electrode surface, e.g., gold through the alkanethiol linker		
Affinity interaction	specific interactions such as those between biotin and avidin or streptavidin	appreciable orientation, stable, well-ordered layer, high functionalisation through specificity and sufficient control	expensive biocompatible linkers and sticky ends modifications

## Data Availability

It is review and it used already publicaly accessible resources/data.

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
