# Peer review of "Design Strategies for Electrochemical Aptasensors for Cancer Diagnostic Devices"

_sensors, 2021, doi:10.3390/s21030736_

Round 1

Reviewer 1 Report

The review “Design strategies for electrochemical aptasensors for cancer diagnostic devices” is dedicated to the analysis of literature data about materials application in electrochemical biosensing. Authors make a hypothesis about the combination of the effects of nanostructures and phys-chemical effects of biomolecules. In review, all pioneering for the interfaces and significant analytes are presented. The several metal states such as bare surface, nanostructures, and clusters for the signal response were taken into account. For the aptasensor interface formation on the nanostructured surface, the pH, solvent, deposition technique (LBL formation, etc.) were discussed. All these parameters influence the final analytical response. The manuscript discusses the practical aspects of using aptasensors for large-scale production. Authors make a conclusion about enthusiastic possibilities of transfer from electroanalytical methodologies to clinical analysis. This is a well-organized review and can be accepted after minor revisions noted:

  • Please give in the end and in each chapter conclusions about comparisons of described techniques in terms of upscaling and technology

Author Response

The review “Design strategies for electrochemical aptasensors for cancer diagnostic devices” is dedicated to the analysis of literature data about materials application in electrochemical biosensing. Authors make a hypothesis about the combination of the effects of nanostructures and phys-chemical effects of biomolecules. In review, all pioneering for the interfaces and significant analytes are presented. The several metal states such as bare surface, nanostructures, and clusters for the signal response were taken into account. For the aptasensor interface formation on the nanostructured surface, the pH, solvent, deposition technique (LBL formation, etc.) were discussed. All these parameters influence the final analytical response. The manuscript discusses the practical aspects of using aptasensors for large-scale production. Authors make a conclusion about enthusiastic possibilities of transfer from electroanalytical methodologies to clinical analysis. This is a well-organized review and can be accepted after minor revisions noted:

  • Please give in the end and in each chapter conclusions about comparisons of described techniques in terms of upscaling and technology

Reply: Thank you. We found it problematic to summarize comparisons on upscaling and technologies in the end of each section: they are somehow already made over the  text and in the Tables, and may be redundant. The focus of this review was more on recent examples, and unfortunately, in our opinion, many of the currently emerging in literature examples are not so useful for direct practical applications and they hardly will be used in clinical practice, for a very common general reason. We prefer not to criticize each particular assay, and introduced a more general summarizing comparison in conclusions and perspectives:

“However, despite numerous scientific reports on excellently in-lab performing electrochemical aptasensors overviewed here and their apparent analytical and technological advantages, the number of practical applications of aptamers in clinical trials, such as for validation of liquid biopsies for diagnosis of cancers or for routine clinical quantification of already established serum biomarkers of cancer, is scarce….. Insufficient long-term stability of bioelectronics chips, such as based on thiolated apatmer immobilisations on gold electrodes, complicated, low-compatible with automatic production electrode modifications with multiple nanomaterials and cumbersome assay protocols make upscaling of most of the suggested technological solutions unfeasible, and they do not approach yet the level of commercial kits for research and development, not mentioning the clinically validated device level. To now, the most successful electrochemical platform for analysis of blood-circulating proteins is Abbott’s iStat Systems for acute disease monitoring and Roche cobas® 6000 ECL analyser for tumor bimarkers, both exploiting the electrochemical ELISA, i.e. the immunoassay approach [52, 280]. The next step may be the development of a similarly commercially-attractive electrochemical aptamer-based platform for unambiguous analysis of protein signatures of the cancer type and cancer state.”

Reviewer 2 Report

The review by Kamila Malecka et al. Design strategies for electrochemical aptasensors for cancer diagnostic devices. Overall the quality of the paper is good. I would like to recommend the review for publication in Sensors after the authors have addressed my minor concerns given below.

  1. Please carefully check for errors between numbers and units throughout the text, such as “2.62 × 10 -15 (0.1 pg mL -1 )” at Page 14, line 549.
  2. Please keep the tables in a consistent format.
  3. Multiplex electrochemical aptasensor platforms and strategies reported for the multiplex analysis (Section 4) need more detailed instructions.

Author Response

The review by Kamila Malecka et al. Design strategies for electrochemical aptasensors for cancer diagnostic devices. Overall the quality of the paper is good. I would like to recommend the review for publication in Sensors after the authors have addressed my minor concerns given below.

  1. Please carefully check for errors between numbers and units throughout the text, such as “2.62 × 10 -15 (0.1 pg mL -1 )” at Page 14, line 549.

Reply: Thank you, done.

  1. Please keep the tables in a consistent format.

Reply: Thank you, done. In the tables, we now use abbreviations only, not both the term and abbreviation, and all abbreviations used in the tables are now translated at the bottom of each table. Also, LODs given in g/mL where relevant were also recalculated in M. We highlighted all changes in yellow, in the revised-highlighted version.

  1. Multiplex electrochemical aptasensor platforms and strategies reported for the multiplex analysis (Section 4) need more detailed instructions.

Reply: We have provided instructions that are more detailed and marked them in yellow.

“5. Multiplex electrochemical aptasensor platforms for several cancer biomarker detection

A simultaneous detection of multiple biomarkers allows efficient screening, diagnosis and monitoring of neoplastic diseases. Despite significant recent advances in the development of aptasensors, most research is still focused on a single analyte detection. However, clinical trials request the accurate and sensitive multianalyte detection, providing a much more detailed information on a range of biomarkers and cancer state.

To address the challenge of saving the time and cost of multiplex analysis, a voltammetric dual sensor for the parallel and continuous detection of MUC1 and CEA was constructed by using a thiolated SH-DNA probe attached to the MCH-blocked gold electrode surface [272]. The MB-labelled MUC1-specific and CEA-specific aptamer sequences were immobilised on the DNA-modified electrodes via their hybridisation with the SH-DNA probe in a way that MB was a signal-on probe for MUC1 binding and a signal-off probe for the CEA detection. The simultaneous analysis of two proteins showed the linear response to MUC1 within the 10 nM to 100 nM range (LOD of 0.6 nM) and from 30 ng mL-1 to 300 ng mL-1 CEA (LOD of 2.75 ng mL-1). Inverse trends in the current signal changes were observed with the increasing concentrations of MUC1 and CEA, generating an increment and a decrement in the current response, respectively. Tumour necrosis factor-α (TNF-α) and thrombin produced significantly lower responses. The aptasensor continuously monitored CEA (after the MUC1 binding) at LOD lowered then to 0.5 ng mL-1 [272]. Moreover, the aptasensor can be regenerated by rehybridization of DNA2 or DNA3-MB after the parallel detection targets, or by introducing MUC1 and CEA and then rehybridizing the MUC1 and CEA aptamers with linker DNA1 on the gold electrode. Therefore, it can be elucidated that this present strategy provides different protocols (parallel and continuous detection) to meet the requirements of analytical applications [272].

Another strategy is based on using multiple redox labels for simultaneous analysis of several proteins, such as an aptasensor exploiting two different integrated signalling probes (ISP) – two aptamers labelled with MB and Fc redox tags (sP1 and sP2) [273]. ISPs were immobilised onto the AuNP-modified GCE (blocked with MCH), and CEA and MUC1 binding resulted in a gradual decrease of both currents from MB and Fc detected by SWV. The linear range for the CEA detection was from 1 ng mL-1 to 1 µg mL-1 and for MUC1 - from 5 nM to 1 µM. LODs for individual analysis of CEA and MUC1 were 0.517 ng mL-1 and 1.06 nM, while for parallel analysis they were 0.5274 ng mL-1 and 1.82 nM, respectively. HSA and HIgG did not interfere with CEA and MUC1 analysis. Comparing with the conventional independent signal probes for the simultaneous multi-analyte detection, the proposed ISP was more reproducible and accurate. This can be due to that ISP in one DNA structure can ensure the completely same modification condition and an equal stoichiometric ratio between sP1 and sP2, and furthermore the cross interference between sP1 and sP2 can be successfully prevented by regulating the complementary position of sP1 and sP2 [273].

Shell-encoded AuNPs: Au@Cu2O core-shell NPs and Au@Ag core-shell NPs were suggested as dual-analysis labels with redox potentials at -0.08 V and 0.26 V, correspondingly [274]. Two aminated DNA probes were immobilised via sulfamine bonds on the surface of GCE containing sulfonic groups introduced by reductive electrolysis of 4-aminobenzenesulfonic acid, and AuNPs-Ag- and AuNPs-Cu2O-conjugated AFP and CEA aptamers were hybridised to both. AFP- and CEA-aptamer complex formations removed the AuNPs-labelled aptamers from the electrode surface, which resulted in the decreasing DPV signals from the shell-encoded AuNPs. The non-interfering and amplified DPV responses enable shell-encoded Au NPs to be an alternative electrochemical signal amplifier for dual screening of CEA and AFP. The duo-sensor showed linearity of responses for both biomarkers in the ranges from 1 pg mL-1 to 10 ng mL-1 AFP and from 5 pg mL-1 to 10 ng mL-1 CEA, with LODs of 1.8 pg mL-1 CEA and 0.3 pg mL-1 AFP. This aptasensor operated in blood samples and cellular extracts, with no interference from PSA, thrombin, IgG, L-cysteine, glutathione, arginine, histidine, valine, tryptophan and lysine. In comparison to the parallel single-analyte assays, shell-encoded Au NPs engineered electrochemical aptasensors offer multiplexing capability and show significant prospects in biomedical research and early diagnosis of diseases [274].

Simultaneous multiplexed detection of CEA and neuron specific enolase (NSE) was performed in a paper-based device produced by wax- and screen-printing, which enabled both the filtration and automatic injection of samples [275]. The device consisted of two integrated working carbon electrodes, which could simultaneously detect CEA and NSE in one sample by using two different redox modifications of the electrode. Each of the electrodes was modified either with amino-functionalised graphene-thionine-AuNPs or Prussian blue–PEDOT–AuNPs nanocomposites, to which either a thiolated CEA aptamer or a thiolated NSE aptamers were attached through the Au-S chemistry. CEA and NSE binding by the aptamers resulted in the linear drop of the DPV signals, at −0.25 V for thionine and at -0.02 V for Prussian blue, within the 0.01 to 500 ng mL-1 CEA and 0.05–500 ng mL-1 NSE concentration ranges. LODs of 2 pg mL-1 CEA and 10 pg mL-1 NSE were orders of magnitude lower than the cut-off serum values for lung cancer: 5 ng mL-1 CEA and 15 ng mL-1 NSE. A label-free electrochemical method was adopted, enabling a rapid simple point-of-care testing. The aptasensor device was used for clinical serum samples analysis, and BSA, uric and ascorbic acids, and Cyfra21-1 did not interfere [275].

Hyperbranched DNA – LaMnO3  perovskite (DNA–LMO) nanobiocomposites loaded with three different metal ions were used as redox probes for simultaneous discrimination of three different biomarkers of cancer, AFP, CEA and PSA (Figure 8, left panel) [276]. Three aptamers were attached to a gold stirring rod and hybridised to DNA-(LMO-M)n encoded probes. The encoded probes were prepared stepwise: 1). Three DNA strands complementary to the CEA, AFP, and PSA aptamers were extended by the TdT enzyme to form a long adenine (A)-rich ssDNA; 2). Pb2+, Cd2+ and Cu2+ ion were loaded onto the LMO nanoparticles then labelled with thymine (T)-rich ssDNA;  3). A-rich ssDNA hybridised with multiple DNA labelled LMO-M to get a set of encoded signal labels corresponding to each biomarker assay DNA-(LMO-M)n. After protein binding, the corresponding encoded labels were substituted by AFP, CEA and PSA biomarkers, and SWV analysis of the supernatant detected the specific metal ion signals from each label encoding the particular protein in one run. The current response linearly increased with the protein concentrations in the range of 0.0001 to 100 ng mL-1, and LODs were 0.34 pg mL-1 AFP, 0.36 pg mL-1 CEA and 0.28 ng mL-1 PSA. Since the stirring rod can enrich many encoded probes containing a lot of metal ions, multiplex signal amplification can be realised. Due to the enrichment and easy separation of the stirring rod, the signal-to-noise ratio was also obviously improved and thus to results in good sensitivity and accuracy. The results of electroanalysis of patients’ sera samples were comparable with those of ELISA [276].”

Reviewer 3 Report

The review described up to date development of electrochemical aptamer based biosensors for detection various cancer markers. It is based on systematic description of fabrication and properties of the biosensors for detection of most occurred cancer markers. The review is well prepared and is suitable for publication in Sensors after minor improvements.

Line 70: Please provide references concerning dimer aptamers and explain their advantage over regular single stranded aptamers.

Line 140: Note that avidin, streptavidin or neutravidin are symmetrical molecules and due to adsorption at the surface not all sites are available for binding of biotinylated DNA molecules. Therefore typically 2 and not 4 sites can be occupied by aptamers. This was described in several papers on acoustic biosensors based on biotinylated aptamers and neutravidin.

Figure 4A - I would recommend to improve scheme of immobilized aptamer as a molecule containing specific binding sites. At the original figure presented it looks like ssDNA without specific binding site.

Author Response

The review described up to date development of electrochemical aptamer based biosensors for detection various cancer markers. It is based on systematic description of fabrication and properties of the biosensors for detection of most occurred cancer markers. The review is well prepared and is suitable for publication in Sensors after minor improvements.

Comment 1: Line 70: Please provide references concerning dimer aptamers and explain their advantage over regular single stranded aptamers.

Reply: Thank you, the text was revised:

“The aptamers can be also readily engineered into bi-specific dimer aptamers [14,15]. Construction of “multivalent” aptamers, dimers in their simplest form, capable of binding to multiple protein binding sites, essentially improves the biorecognition and affinity properties of the aptamers.”

Comment 2: Line 140: Note that avidin, streptavidin or neutravidin are symmetrical molecules and due to adsorption at the surface not all sites are available for binding of biotinylated DNA molecules. Therefore typically 2 and not 4 sites can be occupied by aptamers. This was described in several papers on acoustic biosensors based on biotinylated aptamers and neutravidin.

Reply: We absolutely agree, we meant just a theoretical possibility, sorry for misleading the reader. We correspondingly revised the statement:

“Each streptavidin and neutravidin molecules, possessing four biotin-binding sites, can bind up to two biotinylated aptamers (not all binding sites are accessible for reaction due to steric restrictions induced by these proteins adsorption on electrodes), which increases the number of aptamers on the sensor surface.”

Comment 3: Figure 4A - I would recommend to improve scheme of immobilized aptamer as a molecule containing specific binding sites. At the original figure presented it looks like ssDNA without specific binding site.

Reply: The scheme is reproduced with permission from the original work as it was, we cannot change it by imntorducing new features, but we have added the comment in the figure caption:

Figure 4. (A) Schematic representation of the electrocatalytic aptamer/PEG-based assay for HER-2/neu and voltammograms recorded with the aptamer/PEG-modified electrode in (1) MB and (2) MB and K3[Fe(CN)6] solutions, scan rate 0.1 V s-1 (inset: 5 V s-1) [79]. Specific binding sites and changes of the aptamer conformation upon binding are not shown.”

Reviewer 4 Report

The evaluated paper is a review of electrochemical biosensors for the determination of cancer markers. The scope of the review is limited to biosensors using aptamers. Over 250 references are reviewed. Strategies for the design of biosensors for the ten most frequently used cancer markers are analysed. Brief information on each of the ten cancer markers is given, including cut-off values for most of the markers. Thus, the analytical characteristics (LOD) of biosensors can be compared with diagnostic requirements. Unfortunately, the cut-off values for Platelet-derived growth factors, Mucin 1, Osteopontin and Urokinase Plasminogen Activator are not given. Generally, the review is a significant contribution to the development of biosensors. I recommend accepting the paper with minor revision: the cut-off values for the above-mentioned biomarkers should be provided.

Author Response

Comment 1: The evaluated paper is a review of electrochemical biosensors for the determination of cancer markers. The scope of the review is limited to biosensors using aptamers. Over 250 references are reviewed. Strategies for the design of biosensors for the ten most frequently used cancer markers are analysed. Brief information on each of the ten cancer markers is given, including cut-off values for most of the markers. Thus, the analytical characteristics (LOD) of biosensors can be compared with diagnostic requirements. Unfortunately, the cut-off values for Platelet-derived growth factors, Mucin 1, Osteopontin and Urokinase Plasminogen Activator are not given. Generally, the review is a significant contribution to the development of biosensors. I recommend accepting the paper with minor revision: the cut-off values for the above-mentioned biomarkers should be provided.

Reply: Thank you, we have provided the missing information in the relevant places of the manuscript, more specifically, the cut-offs are:

2062,34 pg/ml cut-off values for Platelet-derived growth factors (Arya, B.; Yamakuchi, M.; Shimizu, T.; Kadono, Jun.; Furoi, A.; Gejima, K.; Komokata, T.; Koriyama, C.; Hashiguchi, T.; Imoto, Y. Predictive Value of Diminished Serum PDGF-BB after Curative Resection of Hepatocellular Cancer. J. Oncol. 2019, ID 1925315).  

0.89 ng/ml cut off value for Mucin 1 (Bastu, E.; Mutlu, M.F.; Yasa, C.; Dural, O.; Nehir, A.A.; Celik, C.; Buyru, F.; Yeh, J. Role of Mucin 1 and Glycodelin A in recurrent implantation failure. Fertil Steril. 2015, 103(4), 1059-1064.

0,538 ng/ml cut off value for Osteopontin (Primasari, M.; Soehartati A. Gondhowiardjo, A.; Handjari, D.R.; Matondang, S.; Kantaatmadja, A.B. Osteopontin level correlates negatively with tumor shrinkage in neoadjuvant chemoradiation of locally advanced rectal cancer. Adv. Mod. Oncol. Res. 2015, 1(1).

0,84 ng/ml cut off value for Urokinase Plasminogen Activator (Hofmann, R.; Lehmer, A.; Hartung, R.; Robrecht, C.; Buresch, M.; Grothe, F. Prognostic value of urokinase plasminogen activator and plasminogen activator inhibitor-1 in renal cell cancer. J Urol. 1996, 155(3), 858-62.

They were also recalculated in M.

Reviewer 5 Report

This is a nice comprehensive review on electrochemical aptasensors. I have following suggestions to make the manuscript attract more readerships.

  1. The authors should also include some fundamental concepts on other transduction techniques although the review is based on electrochemical method.
  2. I suggest the authors to cover in detail the different surface chemistries to develop/fabricate aptasensors.
  3. it would be nice to have a section on different electrochemical techniques as well such as EIS, CV, CA, FETs, etc.
  4. A table summary on the use of nanometerials like CNTs, graphene, CPs and their comparison on detection range, LOD, sensitivity, etc should be provided.
  5. The review should update on the application of aptasensors in real world application examples compared to the lab based sensors.
  6. The review should also talk about the commercialised examples of aptasensors if any...examples of minitiarised aptasensors prototypes.

I understand the authors have put a lot of effort in bringing this review but it is more categorised based on different cancer biomarkers but failed to provide basic fundamental concepts. I would reconsider the manuscript after all these revisions.

Author Response

This is a nice comprehensive review on electrochemical aptasensors. I have following suggestions to make the manuscript attract more readerships.

  1. The authors should also include some fundamental concepts on other transduction techniques although the review is based on electrochemical method.

Reply: To address it, we have added the following material:

“All biosensors can be classified according to the type of a signal transmitted, type of a transducer, and are divided into electrochemical, acoustic, optical and thermal/calorimetric biosensors [44,45]. Thermal biosensors measure the changes in temperature in the reaction between, for example, an enzyme and its analyte substrate. This change in temperature is then correlated with the amount of reactants consumed or products formed [46]. Resonant and acoustic wave biosensors operate by analyzing such measurand as a modulation in the physical properties of the acoustic wave that can then be correlated with the amount of the adsorbed analyte. These devices can also be miniaturized with advantages in terms of size, scalability and cost and easily integrated with microfluidics and electronics for multiplex detection across arrays of numerous devices implanted in a single chip [47]. Despite numerous reports, they have not yet found applications in clinical practice.

Optical detection biosensors are the most diverse class of biosensors exploiting many different spectroscopy techniques, such as UV-vis absorption, phosphorescence, luminescence or fluorescence. Of those, the most established, commercial optical approach for protein analysis is the enzyme-linked immunosorbent assay: ELISA [48,49]. ELISA kits for specific protein cancer biomarkers are validated and widely used in clinical diagnostics of cancer; however, in some cases, they do not provide the sought sensitivity or specificity of analysis, which stimulates the search of advanced biosensor approaches [50]. Another popular type of optical biosensors are based on surface plasmon resonance and use the evanescent-wave phenomenon to describe interactions between receptors immobilized on the biosensor surface and ligands in solution. Binding of the analyte proteins by the surface-immobilized receptors alters the refractive index of the medium near the surface and this change is measured in real time to accurately estimate the amount of bound analyte, its affinity for the receptor and the dissociation and association kinetics of the interaction [51]. Despite many commercial devices offer at the markets, dues to the cost and often insufficient sensitivity they are not yet used in clinical practice.

Electrochemical transducers are numerously used in biosensor research for the detection of cancer biomarkers due to the advantages of easy production, cost-effectiveness, and user friendliness [44,49]. Electrochemical biosensors evaluate the current or potential response produced either due to the analyte binding (capacitive changes) or in the course of an oxidation and reduction reaction at the electrode surface. Of those, electrochemiluminescence (ECL) combining electrogenerated chemiluminescent signal amplification with an optical read out allows improving the sensitivities immunoassays and is currently used for clinical antibody-based analysis of a number of important analytes, including tumor biomarkers. Commercially available ECL analyzers, among those is Roche cobas® 6000, are quite efficient with their 170 to 2170 test h-1 [52], however may not be suitable for direct POCT. Similarly to optical ELISAs, they need a complex equipment for assay performance. Nevertheless, electrochemical detection schemes allow easy miniaturization of the devices and production of portable devices, and here we will consider only this type of transducers. “

  1. I suggest the authors to cover in detail the different surface chemistries to develop/fabricate aptasensors.

Reply: In addition to the immobilization strategies presented in p. 4 and in figure 3, we have added the following material:

“Chemisorption, and above all covalent bonding, as well as chemical affinity are promising methods of aptamers deposition due to the possibility of its attachment to the electrode surface at one point. However, the process of creating a covalent bond is quite complex, it requires modification with appropriate functional groups of the electrode surface, aptamer sequence or both. Table 1 summarises and compares the methods of aptamers deposition on solid surfaces [76,77].

Table 1. Comparison of the methods of aptamers deposition on solid surfaces [76,77].

Immobilisation

strategy

Type of interaction

Advantages

Disadvantages

Electrostatic adsorption

negatively charged aptamer on the positively charged surface

·   fast,

·   simple,

·   low cost method

·  low packing density and control of orientation,

·  random orientation of the aptamer causing layer instability under various conditions, such as changing the ionic strength of the buffer, pH, or other reagents,

·  non-specific adsorption,

·  low binding capacity and operational stability

Covalent attachment

EDC/NHS coupling between COOH- functionalised electrode surface and the amine-terminated aptamer sequence

·   stability (bond can be broken under extreme conditions),

·   well-ordered layer,

·   high degree of orientation control and thickness of electrode surface,

·   single-point attachment of the probe at the end-point thiol group

·   difficult preparation method,

·   high cost

Chemisorption

Involves chemical bond between the probe and the electrode surface e.g. gold through the alkanethiol linker

Affinity

interaction

Specific interactions such as those between biotin and avidin or streptavidin

·  Appreciable orientation,

·  Stable, well-ordered layer,

·  high functionalisation through specificity and well controlled

·    expensive biocompatible linkers and ends up

  1. it would be nice to have a section on different electrochemical techniques as well such as EIS, CV, CA, FETs, etc.

Reply: Thank you. We have provided at as a new Section 2: Electrochemical techniques:

2. Electrochemical techniques

In the electrochemical biosensor approach scrutinized further, the electrode is a transducer element. Electrical changes induced by the protein binding to the biorecognition interface can result either from to reduction/oxidation reactions of a redox marker or due to interfacial changes and thus are analysed in several ways, such as via measuring the current or potential responses when processes involving production or consumption of electrons are involved. When such changes are not due to direct electron flow, the measured responses are resistance, capacitance or impedance. Generally, converting a biological interaction into an electrical signal is straightforward and can be measured and quantified by a variety of methods such as potentiometry, amperometry, voltammetry, conductometry, and impedance [33].

In chronoamperometry, most commonly used in biosensors, direct current is measured by applying a constant potential to the bio-modified working electrode. The amperometric response changes after the analyte binding and that allows to quantify it [34]; the response is produced either by the redox indicator or the analyte itself, if it is electroactive or conditionally can undergo a redox transfomration [34]. The current response is a measure of the electron transfer rate and is proportional to the concentration of the analyte [35,36].

Voltammetric methods, such as cyclic (CV), pulse differential (DPV), alternating current (ACV), linear (LSV), and stripping voltammetry, etc., are another mainstream techniques used in biosensors due to their high sensitivity, precision, accuracy and informativity; they relate the current response from the redox markers to the potential applied [37]. The redox peak potentials are specific to the analysed system, and the magnitude of the peak current is proportional to the concentration of the analyte both if it is redox active or if the proper redox markers are used. Voltammetric techniques are versatile and allow easy extraction of a characteristic information about the analyte. CV is regularily used for determination of formal potentials, redox process mechanisms and electron transfer kinetics [38]. The DPV and square wave voltammetry (SWV) detections become widely applied in biosensors in recent years due to their higher sensitivity, and, as a result, selectivity. In particular, SWV is often used in fast analytical protocols, due to its ability to operate at high frequencies [39], which can also minimize the consumption of electroactive species in comparison to other pulse techniques [40]. The square wave frequency is a parameter that arises from the application of the square wave on the staircase potential and is the frequency at which the analyte is sampled. Similar to CV, the increment in the SWV sweep rate also correlates with an increase in the peak current, however, unlike to CV, this will be proportional to the logarithm of the square wave frequency [41].

Electrochemical impedance spectroscopy (EIS) is another technique heavily exploited in biosensor research, and most productive in detecting the interfacial changes of electrodes functionalized with a biological material [42]. In EIS, a sinusoidal voltage is applied and the resulting current is measured. Impedance is then calculated as a ratio of voltage to current in the frequency domain. During analyte biorecognition and binding, the resistance and capacitance of an electric double-layer change, causing variation in the impedance. By using small amplitude sine wave perturbation, linearity in electrochemical systems can be assumed, allowing the frequency analysis. EIS is further classified as a Faradaic or non-Faradaic EIS depending on the presence or absence of the redox indicator. The second one is more attractive due to no reagents need. Thus, the biorecognition process and label-free interactions on the sensor surface can be detected [36].

Recently, field effect transistors (FET), which are electronic semiconductor-based devices in which current flows are controlled by the applied electric field, started to be populate the biosensor field. Depending on the type of FET system, a p-type or n-type correlating with the type of their charge carriers, either positively or negatively charged analyte species can be sensitively detected by following the change in the conductance response. Novel nanotransducer designs (nanoparticles, nanotubes, nanowires etc.) combined with biomolecule modifications demonstrate impressive sensitivity results, though the stability and sometimes selectivity of sensors may still be an issue [43].

  1. A table summary on the use of nanometerials like CNTs, graphene, CPs and their comparison on detection range, LOD, sensitivity, etc should be provided.

Reply: We are sorry to disagree. The focus of our review is rather based on the analytes, not materials; it is correspondingly structured. We believe that is the most appropriate focus, since the researchers often neglect the main purpose of the biosensor development work: to deliver a practically useful and commercially attractive solution for the specific biomarker detection; they more concentrate on nanomaterial design. Also, it is impossible to compare detection ranges etc. based just on nanomaterial nature: it is far too more complicated problem that lies rather in creating proper affinity/antifouling properties of the surface and application of the corresponding amplification and read out strategy. For different biomarker-aptamer couples it may be very different, and the comparison then should be made as it was done in the current work: for each specific biomarker. The corresponding tabulated data for specific biomarkers are provided in our review.

There were several quite recent reviews focused more on nanomaterials in electrochemical aptasensor design, and we prefer not to follows their footprints: http://dx.doi.org/10.3390/chemosensors8040096; https://www.sciencedirect.com/science/article/abs/pii/S0165993617304430

  1. The review should update on the application of aptasensors in real world application examples compared to the lab based sensors.

Reply: Unfortunately, the examples is scarce, and currently are mostly lab-based, in contrast to electrochemical immunosensors that are commercialised.

  1. The review should also talk about the commercialised examples of aptasensors if any...examples of minitiarised aptasensors prototypes.

Reply: This comment is close to the previous one.We are not aware of any succsessful commercial electrochemical aptasensor for cancer diagnosis. Electrochemical ELISA – yes, and it is now cited in the revised version, but not electrochemical aptasensors:

“To now, the most successful electrochemical platform for analysis of blood-circulating proteins is Abbott’s iStat Systems for acute disease monitoring and Roche cobas® 6000 ECL analysers for tumor bimarkers, both exploiting the electrochemical ELISA, i.e. the immunoassay approach [52, 252].”

Comment 7. I understand the authors have put a lot of effort in bringing this review but it is more categorised based on different cancer biomarkers but failed to provide basic fundamental concepts. I would reconsider the manuscript after all these revisions.

Reply: The focus of this review was detection of different cancer biomarkers indeed, since only few of them are now clinically assessed for cancer diagnosis, and assays are not yet validated in many countries. The review was not eventually focused on basic fundamental aspects, there are too many textbook chapters and other reviews devoted to fundamentals. Our attempt was to overview where we are currently in the electrochemical aptasensor field and how closed suggested solutions are to practical clinical applications. We addressed most of this reviewer suggestions, though. We hope that the revision provided in response to this reviewer comments will not worsen the material presentation by diluting the essential information of biomarker analysis.

Round 2

Reviewer 2 Report

I think the manuscript could be accpted in present form.

Reviewer 5 Report

I recommend the paper for publication.